# Representation of seasonal land use dynamics in SWAT+ for improved assessment of blue and green water consumption

Anna Msigwa[1,2], Celray James Chawanda[2], Hans C. Komakech[1], Albert Nkwasa[2], and Ann van Griensven[2,3]

[1]The Nelson Mandela African Institution of Science and Technology, Arusha 447, Tanzania
[2] Department of Hydrology and Hydraulic Engineering, Vrije Universiteit, Pleinlaan 2 -1050, 1050 Brussel, Belgium
[3] IHE-Delft Institute for Water Education; Westvest 7, 2611 AX Delft, The Netherlands

*Correspondence to*: Anna Msigwa (anna.msigwa@nm-aist.ac.tz)

**Abstract.** In most (sub)-tropical African cultivated regions, more than one cropping season exists following the (one or two)
rainy seasons. An additional cropping season is possible when irrigation is applied during the dry season, which could result in 3 cropping seasons. However, most studies using agro-hydrological models such as Soil and Water Assessment Tool (SWAT) to map blue and green ET do not account for these cropping seasons. Blue ET is a portion of crop evapotranspiration after irrigation application, while green ET is the evapotranspiration resulting from rainfall. In this paper, we derived dynamic and static trajectories from seasonal land use maps to represent the land use dynamics following the major growing seasons to
improve simulated blue and green water consumption from simulated evapotranspiration (ET) in SWAT+. A comparison between the default SWAT+ setup (with static land use representation) and a dynamic SWAT+ model setup (with seasonal land use representation) is made by spatial mapping of the ET results. Additionally, the SWAT+ blue and green ET were compared with the results from the four remote sensing data-based methods, namely: SN (Senay), EK (van Eekelen), Budyko method and Soil Water Balance method (SWB). The results show that ET with seasonal representation is closer to remote
sensing estimates, giving higher performance than ET with static land use representation. The Root Mean Squared Error decreased from 181 to 69 mm/year; the per cent bias decreased from 20% to 13%, and Nash Sutcliffe Efficiency increased from -0.46 to 0.4. Furthermore, the blue and green ET results from the dynamic SWAT+ model were compared to the four remote sensing methods. The results show that the SWAT+ blue and green ET are similar to the van Eekelen method and performed better than the other three remote sensing methods. It is concluded that representation of seasonal land use dynamics
produces better ET results, which provide better estimations of blue and green agricultural water consumption.

## 1. Introduction

Freshwater availability is a limiting resource in many regions worldwide, and the problem is projected to increase in the near future due to land use change, population growth, and climate change. The availability of freshwater is mostly determined by precipitation on land. Rain on land travels via either green or blue
waterways (Velpuri and Senay, 2017; Hoekstra, 2019). The green water resource is the water that is held

in the unsaturated soil layer, whereas the blue water resource is the water that is stored in rivers, streams, surface-water bodies, and groundwater (Falkenmark and Rockström, 2006). One of the solutions to lessen the threat of freshwater scarcity is to minimise consumptive water use in agriculture. However, for water resource management, it is critical to understand water use in agricultural production by source (rainwater or irrigation water from surface and groundwater) (Velpuri and Senay, 2017). Knowing how much direct rainwater (green water) and abstracted water (blue water) is being utilised is crucial for efficient water resource management. Yet such information is not readily available, especially in developing countries.

Hydrological models such as the Soil and Water Assessment Tool (SWAT) can provide information on blue and green water at basin and continental scales (Xie et al., 2020; Jeyrani et al., 2021; Liang et al., 2020; Serur, 2020). For instance, Schuol et al. (2008) used the SWAT model to simulate blue and green water availability for the African continent. Xie et al. (2020) evaluated the evolution of the blue and green water resources, water footprints, and water scarcity in time and space in the Yellow River basin in China from 2010–2018. The study accounts for the effects of irrigation on blue and green water resources. Liang et al. (2020) used the SWAT model combined with future land use and climate scenarios, which was successfully applied to quantify the spatiotemporal distribution of blue and green water change for the Xiangjiang River Basin in China between 2015 and 2050.

However, a few of these studies have implemented annual land use dynamics. Since land use refers to manmade socio-economic activities and management practices on the land, these anthropogenic activities may change depending on the season, specifically on cultivated land (Anderson et al., 1976). These seasonal changes are called seasonal land use dynamics (Msigwa et al., 2019). Hence, mapping the blue and green water with agro-hydrological models such as SWAT needs a better representation of the seasonality/cropping seasons. To the best of our knowledge, no studies have implemented seasonal land use dynamics in the estimation of blue and green water resources. For example, using SWAT, Jeyran et al. (2021) assessed basin blue and green available water components under different management and climatic scenarios. The annual land use change implementation showed that the 30% increase in agricultural land use from 1987 to 2015 has caused significant changes in water shortages in the Tashk-Bakhtegan basin in Iran. However, other studies do not implement even the annual land use dynamic to

decrease the computational time of the large-scale models. In most cases, the dominant soil and land cover are used. For instance, Serur (2020) used a 10-year land use map to model blue and green water availability for the Weyb River basin in Ethiopia.

The main limitation of using these approaches in tropical African cultivated areas is that they typically have more than one growing cycle, usually between 2 and 3, depending on the sequence of rainy and dry seasons and irrigation water availability (Msigwa et al.,2019). Therefore, the right representation and timing of these cropping seasons are important to quantify crop water consumption.

A few studies that have implemented seasonal land use dynamics for other purposes, such as nitrogen leaching and plant growth (Glavan et al., 2015), estimating water withdrawals (Msigwa et al., 2019) and Leaf Area Index (LAI) simulation (Nkwasa et al., 2020), have found an impact of representing seasonal land use dynamics in models. For instance, Nkwasa et al. (2020) found that implementing seasonal land use dynamics in SWAT and SWAT+ models led to an improved vegetation simulation. In addition, the LAI dynamics of the seasonal land use dynamic implementation showed more realistic temporal advancement patterns that corresponded to the seasonal rainfall within the basin. Moreover, Msigwa et al. (2019) found that water withdrawals for irrigated mixed crops increased by 482 $Mm^3$/year when seasonal land use maps are used. On the other hand, seasonal land use dynamics have been studied and evaluated using four methods that use multi-scalar datasets to assess the cropping intensity of smallholder farms. In this study, the cropping intensity is the number of crops planted annually (Jain et al., 2013). However, in this case, the impact of seasonal land use on water resources has not been studied.

The SWAT model incorporates crop rotation and its management at the Hydrological Response Unit (HRU) level within a sub-basin (Neitsch et al., 2002). It is represented as a sequence of planting and harvesting operations within the same HRU supplemented with management operations (Gao et al., 2017). The representation of agricultural management is done through a separate management file, specifying the planting, harvesting, tillage, irrigation, fertiliser, and pesticide application by heat units or month and date (Arnold et al., 2018). Although the SWAT (+) model can represent multiple cropping seasons, this is mainly implemented outside of Africa's catchments. Agro-hydrological model

applications in African basins do not typically represent different cropping seasons. But they implement the default SWAT simulation of a single growing cycle every year (Ndomba et al., 2008; Koch et al., 2012; Gashaw et al., 2018). The lack of consideration of the seasonal land use dynamics in hydrologic modelling studies, particularly in African cultivated basins, may be attributed to past model capability constraints as well as a lack of crop-specific and agricultural management practices data (Van Griensven et al., 2012).

Hence, crop-specific and data management practices could be obtained from the seasonal land use maps using trajectory analysis. Trajectories represent changes in land use over time by comparing changes between two or several land use maps at a grid scale. Trajectory analysis has been widely applied to assess the changes and impact of Land Use and Land Cover (LULC) (Feng et al., 2014; Wang et al., 2012) and as a pre-processing tool for LULC (Zomlot et al., 2017). In these studies, change analysis is done pixel by pixel for each year to identify land use change (Mertens and Lambin, 2000; Swetnam, 2007; Zhou et al., 2008; Wang et al., 2012; Zomlot et al., 2017). However, none of these studies have analysed pixel by pixel within a year to identify the different (cropping) seasons, further referred to as land use dynamics.

A recent study by Nkwasa et al. (2020) in the Usa catchment in the Kikuletwa basin in northern Tanzania has shown how to represent seasonal land use dynamics using trajectories in the SWAT model using the management file and the SWAT+ model using decision tables for accurate hydrological simulation. This study builds on Nkwasa et al.'s (2020) approach to evaluate the effects of seasonal land use dynamics on blue and green ET, with two main objectives: (i) investigate the effect of implementing seasonal land use dynamics on the water balance component in the Kikuletwa basin (6650 km$^2$) with focus on the ET using SWAT+ and (ii) estimate blue and green water consumption from simulated ET.

## 2. Methods

### 2.1. Study Area

The Kikuletwa basin is a sub-basin of the Pangani basin that covers approximately 6,650 km$^2$ (Figure 1). Rainfall within the basin is bimodal, meaning that the area receives long rains (Masika) from March to June and short rains (Vuli) from November to December, as shown in Figure 2. Annual rainfall ranges between 300mm and 800 mm in the lower part of the basin to 1200-2000 mm in the highlands of Mount Meru and Kilimanjaro. The maximum temperature ranges from 25 to 33$^0$C, and the minimum temperature ranges from 15 to 20$^0$C. The basin comprises diverse LULC classes such as agricultural land, dense forest on Mount Kilimanjaro (5880m) and Meru (4562m), grazed land, and mixed urban and shrubland/thickets. Shrubland and thickets in the study area are found mainly in the lowlands, where rain-fed agriculture is dominant. Urban areas concentrate around Arusha, although some emerging small towns are also emerging.

Grazed land is mainly found in the Maasai land of Monduli and Simanjiro districts. Irrigated agriculture in Kikuletwa is mainly practiced in the highlands and lowlands along the river of Moshi, Moshi urban, Hai, Arumeru, Arusha, and Siha districts. The main crops in the highlands are bananas, coffee, and maize, while the lowlands are dominated by mixed vegetable crops such as tomatoes, onions, and beans.

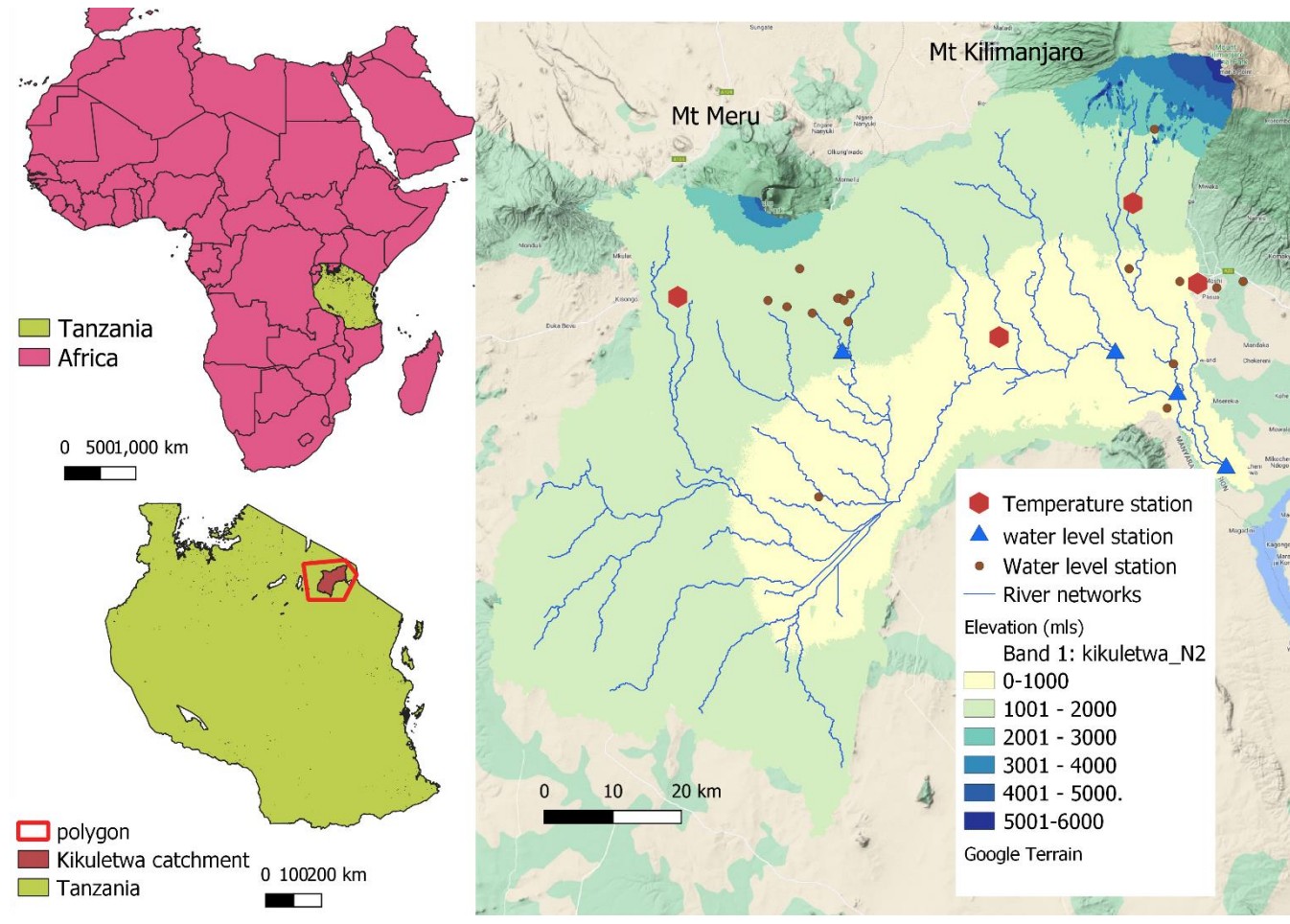

Map data © 2021 Google

**Figure 1.** The location of the Kikuletwa catchment in Africa (inset map). The catchment map shows the river networks and the location of groundwater level, rainfall and temperature station in and around the catchment. (by Authors).

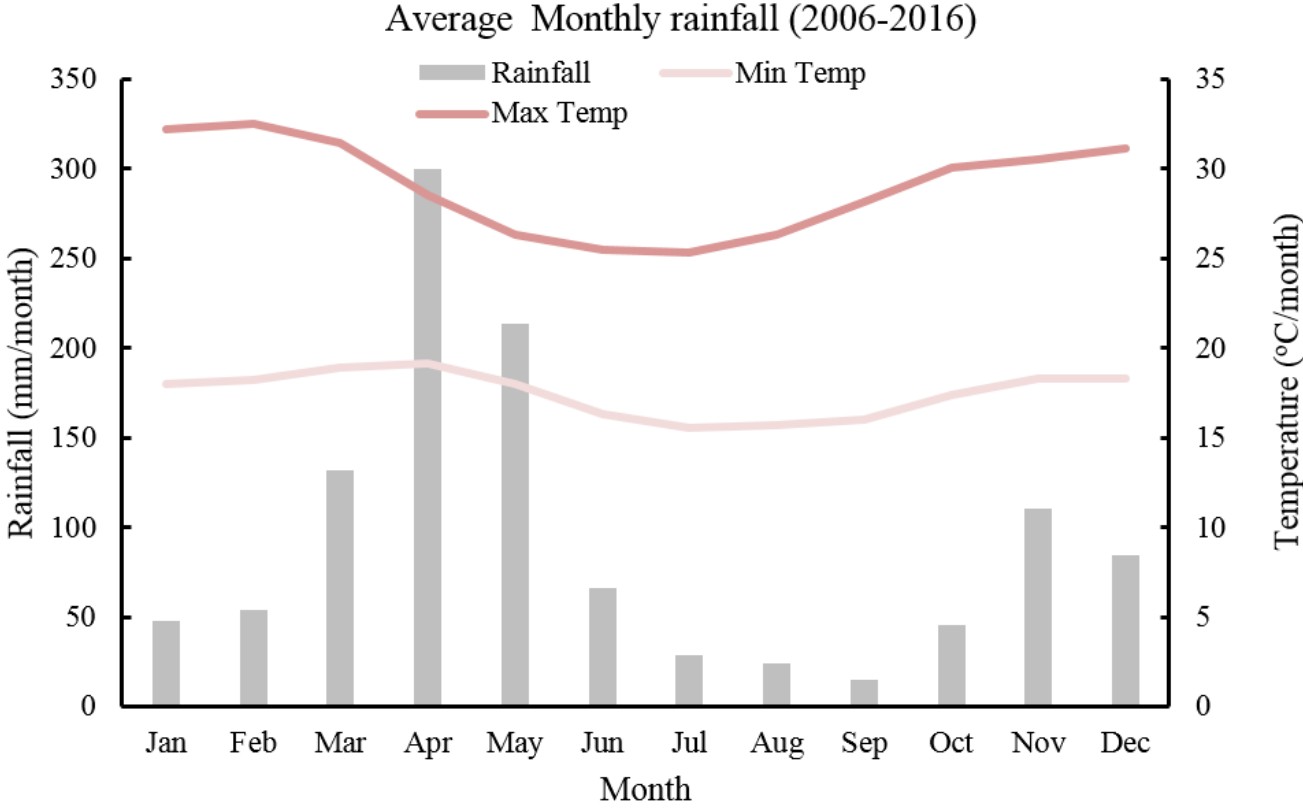

**Figure 2.** Monthly average rainfall (mm) and temperature of Kikuletwa basin ground rainfall stations

## 2.2 Input dataset for SWAT+

The required rainfall, river discharge, climate data, topography, soil map and land use map were collected from different sources. The 90-m Shuttle Radar Topographic Mission (SRTM) Digital Elevation Model

(DEM) was obtained from the United States Geological Survey (USGS) website (https://earthexplorer.usgs.gov/); the soil map was extracted from the African Soil Information Service (AFSIS; Hengl et al., 2015). Daily rainfall records for 10 stations were obtained from the Tanzania Meteorological Agency (TMA) and the Pangani Basin Water Office (PBWO). The daily climate records of temperature (maximum and minimum) for three stations were obtained from PBWO and TMA. The

different data sets had variable record length and quality. However, only good quality data records for the selected 10 rainfall and 4 temperature stations for the overlapping period (2006 to 2013) were selected.

Our study used improved LULC maps with local observation, unlike other studies in the same catchment, such as Notter et al., (2012) and Ndomba et al., (2008). For instance, Notter et al. (2012) used only a few herbaceous crops in model parametrisation without a cropping calendar. The LULC maps were created using Landsat 8 (30 m resolution) images of three months (March, August, and October), representing three seasons in the basin. The March map represents the LULC during the long-wet season (*Masika*), the August map represents the dry season, and the October map represents the short rainy season (*Vuli*). The overall classification accuracy for the land use maps of March, August, and October 2016 were 85.5%, 88.5%, and 91.6%, with a kappa coefficient of 0.84, 0.87, and 0.91, respectively (Msigwa et al., 2019). About 20 and 19 LULC classes in the Kikuletwa catchment were mapped for the wet and dry seasons, respectively. More details on the land use classes and their accuracies are found in Msigwa et al. (2019). The LULC maps were reclassified to match the SWAT land use classification (see Table 3B in Appendix B). For instance, the maps used the SWAT land use code' PAST' to represent grazed grassland.

## 2.3 Land use Trajectories

The LULC change trajectory methodology has been widely applied in many areas to assess LULC change and its impact on the environment. Researchers use trajectories to analyse the change between two images pixel by pixel (Mertens and Lambin, 2000; Swetnam, 2007; Zhou et al., 2008; Wang et al., 2012; Zomlot et al., 2017).

In this study, we extended the meaning of land use trajectories from 'land use change' to 'seasonal succession of land use types for a given sample unit (pixel) with more than two observations at different times' (Zhou et al., 2008). We applied the method in this study to assess the seasonal agricultural dynamics for the meteorological dry and wet seasons of the Kikuletwa basin.

The land use change trajectories were obtained by integrating three classified images to represent the three cropping seasons so that pixel-based change trajectories could be found using GIS. A land use trajectory is the trajectory of a certain pixel in each of the three images. For example, a trajectory of 2→3→0 means that for the given pixel, the land use in March was rain-fed maize (2), then in August,

irrigated mixed crop (3), and finally, in October, bare land (0). This trajectory is classified as dynamic, whereas a trajectory of 4→4→4, meaning the land use is irrigated banana and coffee (4) in March, August, and October, is static. Thus, the LULC change trajectories were categorised into dynamic and static land use trajectories. We only implemented the trajectories from all agricultural land uses except irrigated banana and coffee and irrigated banana, maize and coffee land uses, which were combined as irrigated banana and coffee land use. About 74% of the trajectories were static, while 26% of the trajectories were dynamic. Figure 3 shows the spatial distribution of static and dynamic land use trajectories found in the study area. Only agricultural land use and extensive agriculture LULC, such as grazed grassland and shrubland, were considered when analysing the seasonal changes (dynamic land uses) and implemented in the SWAT+ model. We analysed and implemented 40 land use trajectories, Appendix B; Table 1B shows a few of the implemented trajectories.

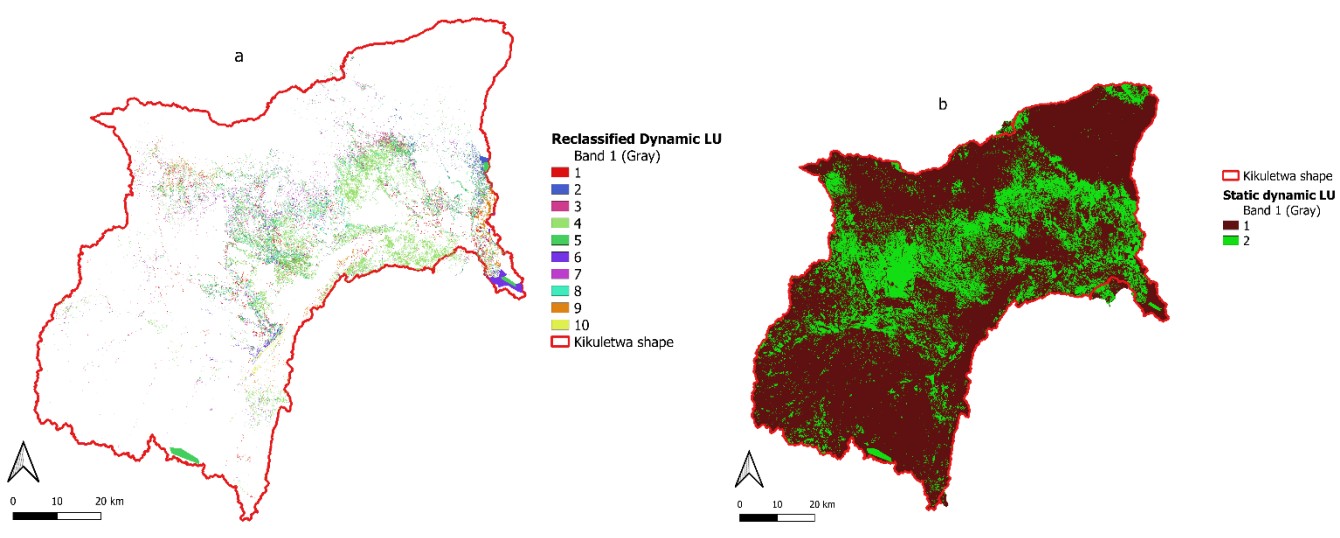

**Figure 3.** Spatial distribution of main dynamic land use trajectories (a) and the distinction between dynamic and static land use (b) identified in the study area.

| Legend | | |
| --- | --- | --- |
| **Id** | **Main Trajectory** | **Crop/ vegetation cover meaning** |
| 1 | AGRL-BSVG-AGRL | Beans-space vegetation-beans |
| 2 | CORN-AGRL-PAST | Rainfed maize-beans-grassland |
| 3 | CORN-AGRL-BSVG | Rainfed maize-beans-space vegetation |
| 4 | AGRL-AGRL-BSVG | Irrigated mixed crops- irrigated mixed crops -space vegetation |
| 5 | CORN-AGRL-AGRL | Rainfed maize- Irrigated mixed crops - Irrigated mixed crops |
| 6 | AGRL-AGRL-AGRL | Irrigated mixed crops - Irrigated mixed crops - Irrigated mixed crops |
| 7 | CORN-PAST-AGRL | Rainfed maize -grassland- Irrigated mixed crops |
| 8 | AGRL-AGRL-PAST | Irrigated mixed crops - Irrigated mixed crops -grassland |
| 9 | SUGC-AGRL-AGRL | Irrigation sugarcane- Irrigated mixed crops - Irrigated mixed crops |
| 10 | AGRL-AGRL-AGRL | Irrigated mixed crops - Irrigated mixed crops - Irrigated mixed crops |

## 2.4. SWAT+ Model

SWAT+ is a physically-based, semi-distributed hydrological model and restructured version of the Soil

and Water Assessment Tool (SWAT) designed to address current and future challenges in water resource modelling and management (Bieger et al., 2017). Due to its watershed discretisation and configuration, SWAT+ is more flexible in simulating basin processes such as evapotranspiration, runoff, crop growth, and nutrient and sediment transport. The HRUs are contiguous areas, i.e., a representative field, with an associated user-defined length and width. The actual HRU is calculated based on the DEM, soil, and land

use map inputs. Subbasins are delineated during the model construction but are divided into water areas and one or more landscape units (LSU) (Bieger et al., 2017).

Land use and management representation in SWAT+ can be done through the management file or using decision tables. Decision tables are an accurate yet compact way to model complex rule sets and their corresponding actions. Nkwasa et al. (2020) highlighted the greater flexibility provided by decision tables

during the representation of agricultural practices in SWAT+. The model gives room for two or more crops growing simultaneously by defining the plant community in the specific plant file. The model enables the representation of the reality of cultivated tropical basins.

The ET in the model is estimated at HRU level. Different methods (Priestley-Taylor, Penman-Monteith, and Hargreaves) are used to estimate ET in the SWAT+ model. More detailed information can be found

in (Abiodun et al., 2017; Alemayehu et al., 2016; Neitsch et al., 2002). Our study adopted the Hargreaves method (Hargreaves and Samani, 1982) to estimate ET due to the limited amount of input data, such as solar radiation. The method has been tested in tropical basins such as the Mara basin, linking Tanzania and Kenya (Alemayehu et al., 2016). We aimed to use available ground data and not rely on remote sensing climate data such as solar radiation, which is reported to have uncertainties (Alemayehu et al., 2016). The SWAT model has also been successfully used in the Pangani basin for different purposes (Ndomba et al., 2008; Notter et al., 2012).

## 2.5 Land use Trajectories Implementation in SWAT+

We combined three maps (March, August, and October) to obtain the trajectory land use map. Forty land use trajectories were produced from the three seasonal land use maps. These trajectories differ from the traditional approach in that they define the space using agricultural static and dynamic land use maps. Then each trajectory was assigned a SWAT+ land use code (placeholder). For instance, a placeholder SWAT+ land use code 'MIXC' signifies a CORN→TOMA→TOMA trajectory (rainfed maize to tomato-to-tomato land use trajectory), or 'MIGS' signifies a CORN →TOMA →BSVG trajectory (rainfed maize to tomato to sparse vegetation land use trajectory) as shown in Table 1B (Appendix B). The trajectory land use map is represented with the placeholder SWAT+ land use codes using the lookup Table 1B (Appendix B) for the Kikuletwa basin was created. A python code (Appendix A) was used to assign trajectories of the placeholder SWAT+ land use codes and to create the trajectories' management files, i.e., the 'landuse.lum', 'management.sch', and 'hru-data.hru' files. In the 'Landuse.lum' file, the trajectories were defined with respect to the plant community. The 'Management.sch' file controls the timing of the planting and harvesting of the individual crops in the community (Table 1). For instance, tomatoes and soya beans are planted in the same field with different planting and harvesting schedules but grown during the same period. However, each crop was defined by its own plant community in the new SWAT+ to distinguish between these crops. The 'hru-data.hru' file links the HRUs to the corresponding land use management. The irrigation schedules were implemented using decision tables. The source of irrigation water in the catchment was the river, and irrigation technique was mostly furrow.

**Table 1.** An example of a 'management.sch' file input in dynamic SWAT+ model

| name | numb_ops[9] | numb_auto[10] | op_typ[11] | Mon[12] | Day[13] | hu_sch[14] | op_data1* | op_data2* | op_data3* |
|---|---|---|---|---|---|---|---|---|---|
| cor_agr_agr_m[1] | 8 | 2 | | | | | | | |
| | | | irr_toma_soy[2] | | | | | | |
| | | | irr_corn[2] | | | | | | |
| | | | plnt[3] | 3 | 15 | 0 | corn[5] | grain[8] | 0 |
| | | | hvkl[4] | 8 | 15 | 0 | corn | grain | 1 |
| | | | plnt | 7 | 1 | 0 | soyb[6] | grain | 2 |
| | | | plnt | 8 | 20 | 0 | toma | null | 3 |
| | | | hvkl | 10 | 1 | 0 | soyb | grain | 4 |
| | | | hvkl | 10 | 20 | 0 | toma[7] | null | 5 |
| | | | plnt | 10 | 30 | 0 | corn | grain | 6 |
| | | | hvkl | 2 | 28 | 0 | corn | grain | 7 |
| agr_agr_agr_m[1] | 8 | 2 | | | | | | | |
| | | | irr_toma_soy[2] | | | | | | |
| | | | irr_corn[2] | | | | | | |
| | | | plnt | 3 | 15 | 0 | soyb | grain | 0 |
| | | | hvkl | 6 | 30 | 0 | soyb | grain | 1 |
| | | | plnt | 7 | 1 | 0 | soyb | grain | 2 |
| | | | plnt | 8 | 20 | 0 | toma | null | 3 |
| | | | hvkl | 10 | 1 | 0 | soyb | grain | 4 |
| | | | hvkl | 10 | 20 | 0 | toma | null | 5 |
| | | | plnt | 10 | 30 | 0 | corn | grain | 6 |
| | | | hvkl | 2 | 28 | 0 | corn | grain | 7 |

[1] name of the land use management, [2] points to the irrigation decision tables, [3] planting operation, [4] harvesting operation, [5] rainfed maize, [6] soy bean, [7] tomato, [8] harvest the grain portion of the crop, [9] number of operations, [10] number of auto-operations, [11] operation type, [12] month, [13] day, [14] heat unit schedule, * operations

## 2.6 Model Configuration for both Static and Dynamic SWAT+ Models

The SWAT+ model was set up using DEM, soil map and land use map of March 2016 for the static representation scenario (static model) and using a trajectory map and files (described in section 2.5) for the dynamic representation scenario (dynamic Model). In the static model, the crops were grown in the rainy season from March to July, and the land would be left bare. This is normally the case with most SWAT model applications in SWAT (Ndomba et al., 2008; Gashaw et al., 2018; Koch et al., 2012). Both models used the same ground observations of rainfall and temperature (Appendix C, Table 1C). The precipitation stations were adjusted manually according to elevation, and the potential maximum leaf area index of maize was adjusted to correspond to the field measurements of the basin. The USDA Soil

Conservation Service (SCS) curve number was used to estimate surface runoff and the Muskingum method was used for channel routing.

For the static SWAT+ model, 23 sub-basins, 171 landscape units, and 6086hru were generated with 14 land use classes, while for the dynamic SWAT+ model, 23 sub-basins, 171 landscape units, and 9333hru were generated with 40 land use classes representing the 40 different trajectories. The difference in the number of HRUs is related to the higher number of land use classes in the dynamic land use mapping. The irrigation schedules were implemented through decision tables (Arnold et al., 2018) by specifying a furrow irrigation method and using the rivers within the sub-basins as the source of irrigation. The model was run for a period of 8 years (2006 to 2013). The first two years were used as a warmup period.

## 2.7 Model Evaluation

Both the static and dynamic SWAT+ models were compared in terms of how they simulate the water balance, with a particular emphasis on the ET component because the primary goal of this study is to improve the spatial distribution of blue and green water consumption. Hence, the SWAT+ models were not calibrated. The ET from both static and dynamic SWAT+ representation scenarios was compared with the remote sensing ET at a basin level for the same simulation period from 2008 to 2013. The remote sensing ET is an ensemble ET product from seven existing global-scale ET products (IHE Delft, 2020). All the ET products are based on multi-spectral satellite measurements and surface energy balance models, i.e., the Global Land Evaporation Amsterdam Model (GLEAM) (Miralles et al., 2011), CSIRO MODIS Reflectance-based Evapotranspiration (CMRS-ET) (Guerschman et al., 2009), Operational Simplified Surface Energy Balance (SSEBop) (Senay et al., 2013), Atmosphere-Land Exchange Inverse Model (ALEXI) (Anderson et al., 2007), Surface Energy Balance System (SEBS) (Su, 2002), ETMonitor (Hu and Lia, 2015) and MODIS Global Terrestrial Evapotranspiration Algorithm (MOD16) (Mu et al., 2011). Detailed information on the ET products' description and method is found in Hugo et al. (2019). The product was evaluated for the study area by comparing the basin water balance at three gauged stations; Karangai, Kikuletwa Power station, and Tanzania Plantation Company (TPC) over a period of six years (2008-2013). The comparison of ET calculated using the water balance and remote sensing

showed good agreement (NSE= 0.77) for Kikuletwa Power station, which covered 86% of the total basin area (Msigwa et al., 2021, 2019). Statistical metrics such as Nash-Sutcliffe efficiency (NSE), Root Mean Square Error (RMSE), Percent Bias (PBIAS) and adjusted R squared ($R^2$) were used to compare monthly ET from static and dynamic SWAT+ models to the remote sensing ET. Moreover, the Paired T-test statistical analysis was performed to determine if there is a significant difference between the ET from 270 the static model and that of the dynamic model for only the dynamic land uses.

## 2.8 Estimating blue and green ET

The blue ET is a portion of crop evapotranspiration after irrigation application, while green ET is the evapotranspiration resulting from rainfall. The blue ET in this study was estimated as a difference between ET under irrigation and ET without irrigation (Liu and Yang, 2010). The SWAT+ dynamic land use 275 implementation was run without irrigation, and then later, irrigation was applied. The green ET is the actual evapotranspiration from precipitation which can be kept in unsaturated soil and absorbed by plants and is then returned to the atmosphere via evapotranspiration. In this study, only the portion of blue water consumed from irrigation was considered and not all the blue water resources like in other studies (Xie et al., 2020).

The SWAT+ model was run first, assuming no irrigation was carried out. The computed ET is called $ET_{green}$. Then the SWAT+ model was run again with irrigation being implemented, and the ET computed is called $ET_{total}$ as explained in the two scenarios below. Finally, $ET_{blue}$ is computed by the difference of $ET_{total}$ from the run with irrigation implantation and $ET_{green}$ "Eq. (4)".

The two scenarios used to estimate blue ET
1. The seasonal dynamic SWAT+ is carried out by assuming the soil does not receive any irrigation water. The evapotranspiration computed using this first run is referred to as $ET_{green}$
   2. The seasonal dynamic SWAT+ is carried out by assuming the soil receives sufficient irrigation water. The evapotranspiration computed using this second run is referred to as $ET_{total}$

Hence, $ET_{blue}$ is computed from the "Eq. (4)" below.

$ET_{blue} = ET_{total} - ET_{green}$        (4)

It should be noted that the trajectory implementation involves only two of the agricultural land uses, i.e., rainfed maize and mixed crop, except for irrigated banana and coffee land use and irrigated banana, coffee and maize land use.

## 2.9 Comparison of SWAT+ results with other remote sensing methods

The SWAT+ blue and green ET were compared with the results from the four remote sensing data-based methods, namely: SN (Senay et al., 2016), EK (van Eekelen et al., 2015), the Budyko method (Simons et al., 2020) and the Soil Water Balance method-SWB (FAO and IHE Delft, 2019).

The SN method (Senay et al., 2016) is the simplest method whereby blue water is estimated as a difference between precipitation (P) and ET, followed by the modified method of van Eekelen et al. (2015) where

the effective fraction was introduced to reduce the amount of precipitation that evaporates. The Budyko method, as described in Simons et al. (2020), estimates green water from precipitation using an empirical relationship between actual evapotranspiration, precipitation, and reference evapotranspiration. The Budyko equation, also called the Budyko curve, assumes a relationship between the evaporation ratio (ET/P) and the climate aridity index (ETo/P) to describe the water-energy balance for long-term analysis.

The soil moisture balance model computes green ($ET_{green}$) and blue ($ET_{blue}$) water components of ET by keeping track of the soil moisture balance and determining whether ET can be satisfied through direct precipitation and precipitation stored as soil moisture alone or if additional water (surface or groundwater supply) is required. The study compares blue and green water estimations for all LULC classes for the Kikuletwa catchment.

## 3. Results

### 3.1 Comparison of Simulated basin ET from Remote Sensing

Figure 4 shows the average monthly ET at the basin scale of Kikuletwa for the two model scenarios of SWAT+ and that from remote sensing. The dynamic SWAT+ model shows higher ET (by 20mm/month), matching the remote sensing pattern in the dry seasons (July to October) than the static SWAT+ model

implementation. This shows that agricultural activities are occurring in the dry seasons. In the dynamic SWAT+ model, we implemented irrigated cropping during the dry seasons, leading to an increase in ET.

The statistical analysis (Table 2) shows that both the SWAT+ simulations have a correlation ($R^2$) of above 0.5 when compared with the monthly remote sensing ET. However, the monthly average ET value for the dynamic land use scenario is closer to the remote sensing ET, especially during the dry months from July to November, where we implement more than one cropping season.

Unlike the commonly used static land use scenario where only one cropping season was implemented per year, the monthly ET for the dynamic SWAT+ model implementation shows an acceptable PBIAS of 13%. In contrast, the static SWAT+ model shows a higher PBIAS of 30%. Moreover, the dynamic SWAT+ model shows a good NSE of 0.4, while the static SWAT+ shows very low performance with an NSE of -0.46.

Table 3 shows the water balance component for the two scenarios. A notable difference is seen in ET increase (24%) and decrease in other water balance components (lateral flow; 27%, percolation; 42%, surface runoff; 32%). In addition, the mass balance (change in soil water balance) in percentage for the static SWAT+ model is higher (1.8%) than in the dynamic SWAT+ model (0.5%). The most pronounced differences are found when comparing the dynamic land use representation on a basin scale and the commonly used static land use approach with remote sensing. Figure 5 shows the spatial distribution of ET from remote sensing, dynamic land use and static land use representation.

The average basin ET is 461mm/y, 573mm/y and 642 mm/y for the static SWAT+ model, dynamic SWAT+ model, and remote sensing, respectively. Generally, all the simulated ET from SWAT+ shows a lower annual average ET than remote sensing ET. However, the ET from static land use representation shows a higher difference of 181mm/y, whereas, with dynamic land use, the difference in ET is only 69mm/y. The paired T-test results show a significant difference between the ET from the static model and that of the dynamic model for the dynamic land uses. A P value of 0.013 was obtained, which was less than the 0.05 confidence interval. The spatial distribution of ET from the SWAT+ models is different

from remote sensing. However, visually, the spatial distribution of ET from the dynamic land use is closer and shows similar patches to remote sensing than the ET from the static land use scenario (Figure 5).

The differences in ET spatial distribution (Figure 5) are vivid mostly in the trajectory implemented areas in the lowlands (see Figure 3). Figure 6 shows the ET on the dynamic land uses alone. The difference in

the values of ET in these areas is more than 100mm per year. The vivid differences are seen in the right lower corner of the catchment, where the differences in ET are more than 200mm/y. There are more areas with less than 400mm/y in the static model compared to the dynamic model.

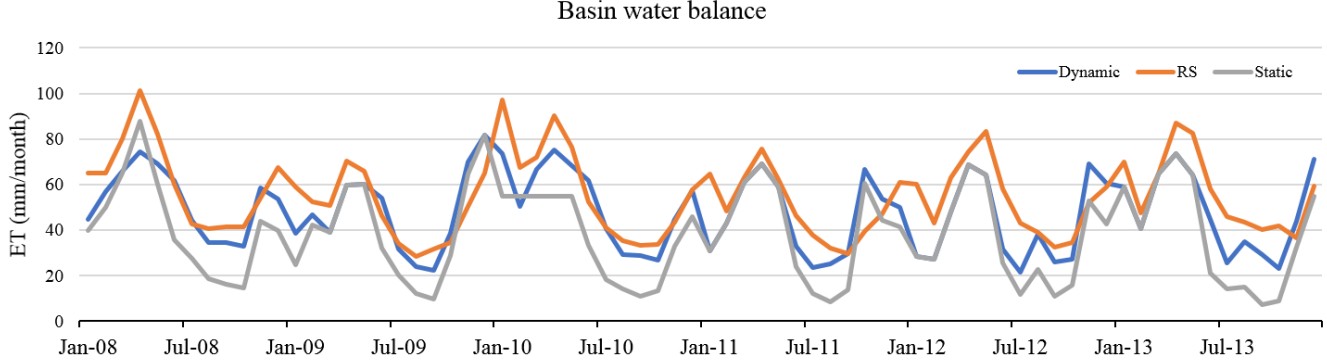

**Figure 4.** Average monthly ET for basin-scale summarised from remote sensing, dynamic land use

scenario and static land use scenario.

**Table 2.** Statistical analysis of ET comparison of SWAT scenarios from Remote sensing

| Statistic Parameter | Static SWAT+ | Dynamic SWAT+ |
|---|---|---|
| PBIAS | 30% | 13% |
| Nash-Sutcliffe efficiency (NSE) | -0.46 | 0.4 |
| Adjusted R Square | 0.6 | 0.6 |
| RMSE (mm/month) | 20.8 | 13.3 |

**Table 3.** Comparison of water balance component for the basin level

| Water balance component (mm) | Static | Dynamic |
|---|---|---|
| Precipitation | 814 | 814 |
| Irrigation | 0 | 8.25 |
| Evapotranspiration | 461 | 573 |
| Lateral flow | 139 | 101 |
| Surface runoff | 207 | 140 |
| Percolation | 21.7 | 12.6 |
| %mass balance | 1.8 | 0.53 |

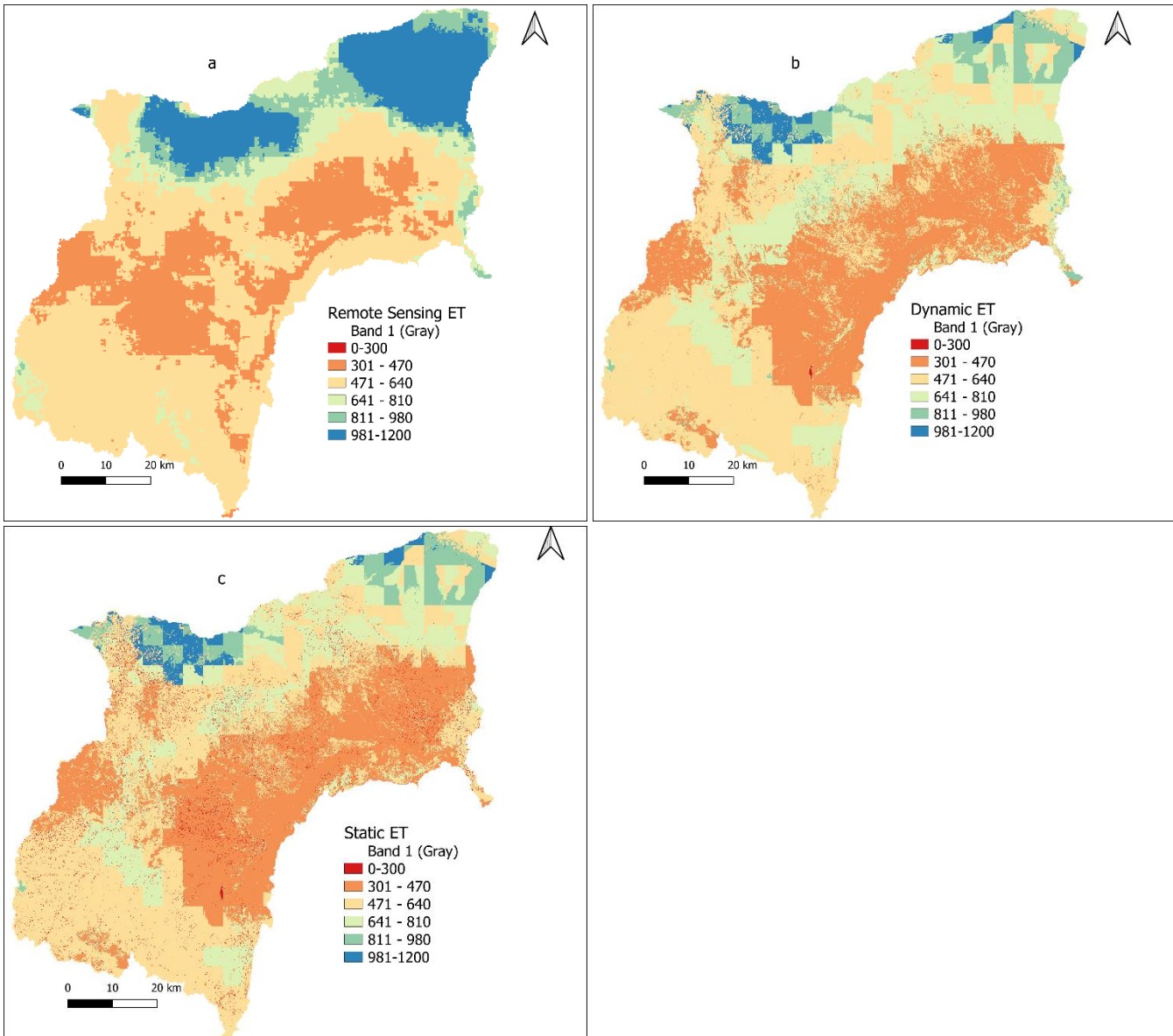

**Figure 5**. Spatial distribution of ET from (a) Remote sensing (b) dynamic land use scenario and (c) static land use scenario.

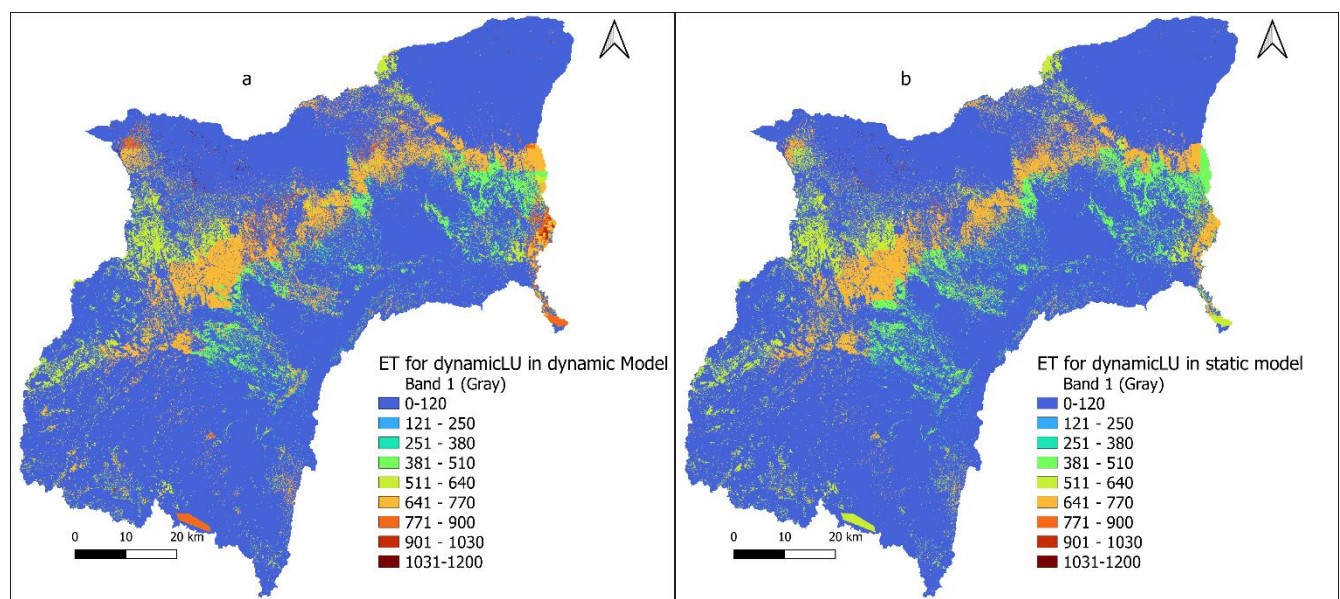

**Figure 6**. Spatial distribution of ET from dynamic Land use for both (a) dynamic and (b) static SWAT+ Models.

## 3.2 Blue and Green ET

Figure 7 shows the blue and green annual ET trends in the Kikuletwa basin from 2008 to 2013. The implemented blue and green ET was mainly for irrigated mixed cropland use due to the implementation of trajectories. The annual average blue ET for irrigated mixed crops is 138mm, which accounts for 25.5% of the annual average total ET, and the annual average green ET is 402mm, which accounts for 74.5% of the annual average total ET.

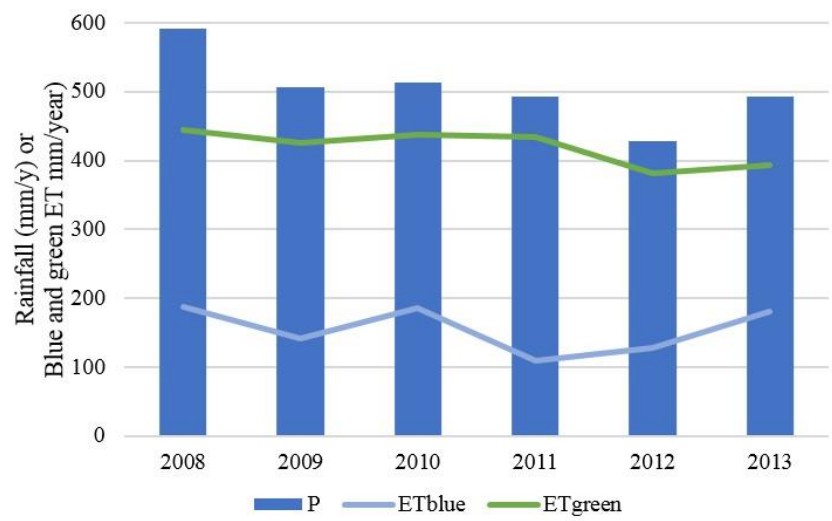

**Figure 7.** The annual variation of blue and green ET from 2008–2013.

Figure 8 shows the spatial distribution of blue ET for agricultural areas in the Kikuletwa basin for
implemented trajectories such as rainfed maize to tomato to irrigated maize land use trajectory (See
Appendix 2, Table 2). The blue water is calculated from the irrigated implemented trajectories that mainly
include irrigated mixed crops (soybeans, tomato, and irrigated maize). Figure 8 shows that more than half
of the total area consumes less than 200mm of blue ET. The higher blue ET is seen in the lower right
corner where the irrigated sugarcane plantation is found.

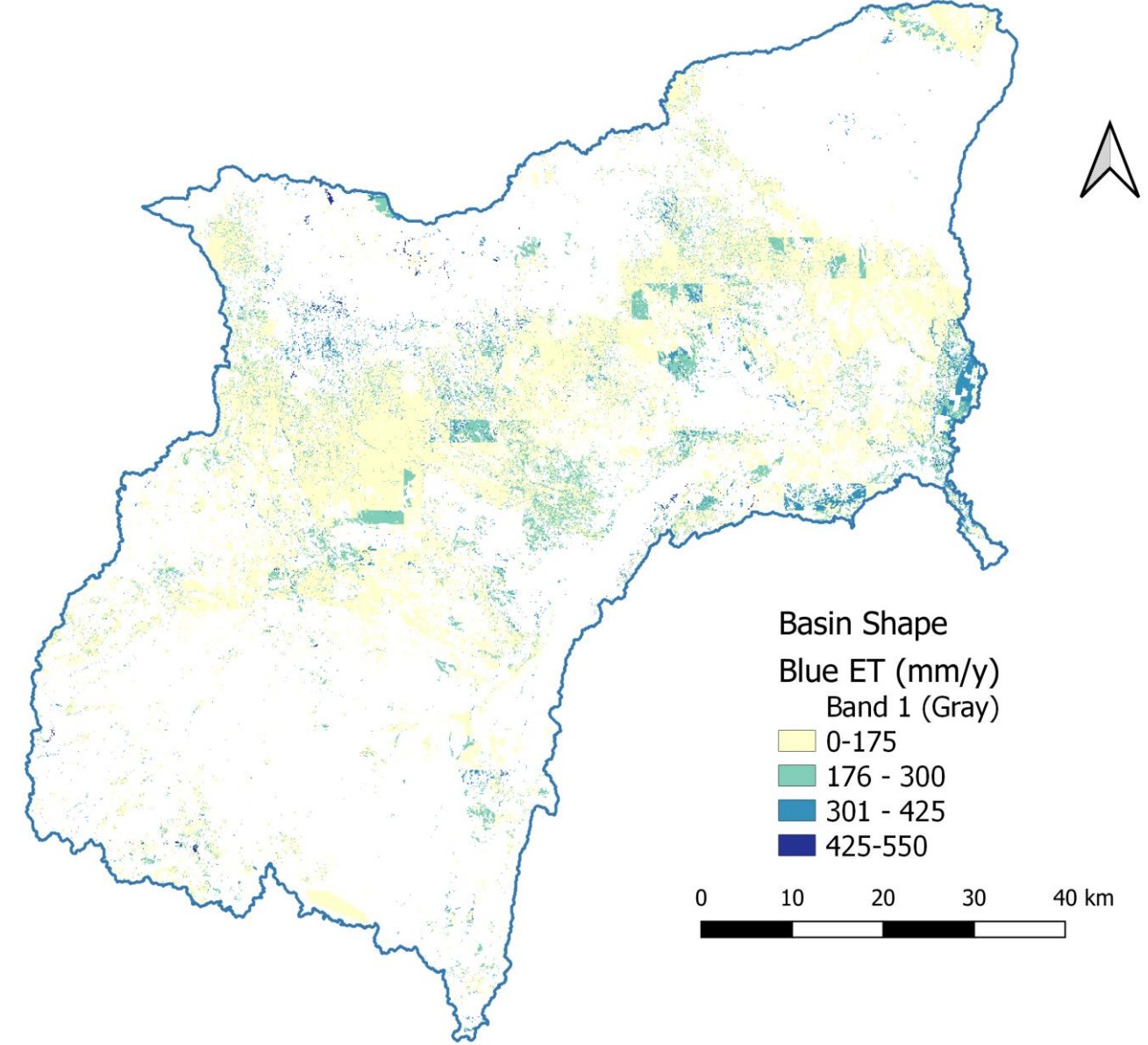

**Figure 8**. Spatial distribution of Blue ET for the implemented trajectories of rainfed and irrigated mixed crops land use.

Figure 9 compares average blue and green ET from four methods (Msigwa et al., 2021) with dynamic SWAT+. The value of both blue and green ET is closer to two methods, the EK (van Eekelen) and SWB (Soil Water Balance) methods, which were indicated to have realistic blue and green ET values. The Van

Eekelen et al. (2015) is the method that analyses precipitation (P) and ET and applies an effective rainfall factor since not all rainfall will infiltrate and be stored in the unsaturated zone to be available for uptake by plants. Both ground data and remote sensing data could be used for data analysis-based approaches on an annual basis. The SWB model is a pixel-by-pixel vertical soil water balance model that splits green and blue ET by tracking soil moisture balance and determining if the ET is satisfied only from rainfall or stored in the soil moisture or additional sources if required (FAO and IHE Delft, 2019).

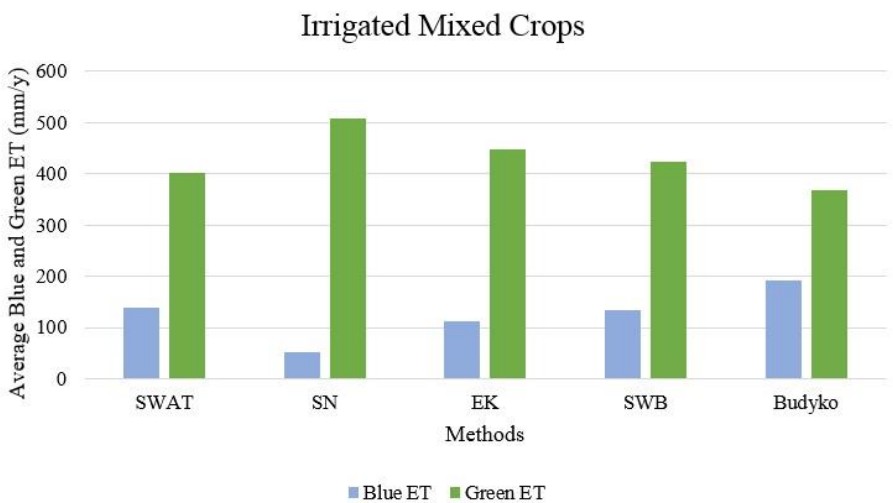

**Figure 9.** Blue and green ET comparison with other four methods from Msigwa et al. (2021).

## 4. Discussion

Previous studies have represented annual land use changes in SWAT and found that these significantly impact hydrology (Wagner et al., 2016; Woldesenbet et al., 2017; Wagner et al., 2019). However, none of these studies has represented the seasonal dynamics of land use within a single year in a spatially distributed manner. Nkwasa et al. (2020) incorporated the seasonal land use dynamic into SWAT and SWAT+ and found that models led to an improved vegetation simulation. This study did not show how

the seasonal land use dynamic improved water balance components such as ET. Our study uses an agro-hydrological model (SWAT+) to represent blue and green ET for different cropping seasons (represented by trajectory with time and space) and the use of remote sensing ET to evaluate the simulated ET from

SWAT+. The study has compared a common default modelling approach where a static land use map is used together with its management practices with a seasonal dynamic land use representation where more than one cropping season is represented in a year. The spatial and temporal ET estimates from two model setups were compared with remote sensing ET. An increase of 112mm/y of the ET is seen when seasonal dynamic land use is implemented in the dynamic model to match the remote sensing ET as compared to when a static land use map is used in the static model. The ET results from the dynamic model are significantly different from the ET in the static model for the dynamic land use. The models show differences in water balance components. This is due to the implementation of the land use trajectory in the dynamic model.

A remarkable difference is seen in the spatial distribution of ET from static and dynamic land use SWAT+ representation. The dynamic land use SWAT+ visually is similar to a remote sensing map compared to the static land use SWAT+. This is because of the added management practices such as irrigated cropping in the dry seasons, unlike the default SWAT+ with static land use throughout the simulation period. The ET from the dynamic land use setup could not reach the maximum satellite ET because the satellite ET estimates also have uncertainties in the mountainous areas because of the presence of cloud cover. Moreover, different methods for estimating ET could lead to these differences. Climate ground stations (temperature, wind speed, relative humidity, and solar radiation) were used for ET simulation in the SWAT+ model, while remote sensing uses energy balance models, mostly remote sensing data.

On the other hand, the ET from the static land cover, such as forest from the static and dynamic model setups, shows different ET values. This could be because of the difference in the initial model setup. The model setup for static used a March land use map with only 14 land use classes, while the dynamic model used a land use map with 40 trajectories. Hence, the changes in the ET might be due to the different land use maps yielding different numbers of HRUs. In order to avoid such a difference, one could have an initial setup with the same land uses, and then trajectory implementation could only be with the agricultural land use.

Furthermore, the ET estimates from the dynamic SWAT+ model were used to estimate blue and green ET. The blue and green ET estimates from SWAT+ for the mixed cropland use show no significant

difference in the values from the two methods (EK and SWB) assessed in Msigwa et al., (2021).

These findings demonstrate the importance of representing seasonal land use dynamics in modelling blue and green water consumption. Normally, most models use NDVI to represent seasonal changes (Amri et al., 2011; Ferreira et al., 2003), whereas the use of dynamic land use leads to improved accuracy of seasonal simulations of water use (Nkwasa et al., 2020). Seasonal land use maps can add information on

management practices of changes in temporal crop rotation and irrigation water use at a spatial scale. However, to account for accurate seasonality of land use, more than three maps within a year should be represented, ideally 12 maps each year. This would enable a more complete understanding of the agricultural land use classes and minimise errors in the trajectory analysis. However, Landsat 8 is associated with clouds, especially in the rainy season. Therefore, cloud masking techniques are needed

before further analysis of the images. Also, there were uncertainties associated with the trajectories; for example, unrealistic trajectories like changing from crop to forest and then to crop again. These types of trajectories were corrected and reclassified.

The Landsat 8 images used in this study to map seasonal land use dynamics did not have a revisit time (16-day) that is small enough to acquire an adequate number of monthly images to represent the year.

More products are now becoming available (Sentinel-2, 5-day revisit time) with a higher temporal resolution, which would aid in collecting more cloud-free images to represent seasonality within the year.

Although it appears important to include seasonal land use dynamics, one may claim that the annual land use implementation is enough when studying the effect of land use in hydrology. Our study shows a significant impact of the representation of seasonal land use in the SWAT+ model by reducing the errors

in water consumption estimations.

## 5. Conclusion

Understanding the spatial-temporal variability of agricultural water consumption in terms of blue water requires accurate estimates of ET. This study has demonstrated the importance of incorporating seasonal land use dynamics to improve simulated ET for further blue and green ET estimates using a SWAT+ model. Although the static representation gives equally good $R^2$ results of more than 0.5, we found that the RMSE for the static model result is significantly higher than the RMSE of the dynamic model result by about 112 mm per year. Moreover, the ET from the dynamic SWAT+ model gave a low PBIAS (13%) and a relatively good NSE of 0.4 compared to the ET from the static SWAT+ model, which gave a higher PBIAS (20.8%) and a negative NSE of -0.46. The study showed that a dynamic land use representation in the SWAT+ model gave ET estimates closer to the remote sensing ET than the default model with a static land use representation. The improved ET map from the dynamic SWAT+ model improved the blue ET estimates as compared to the use of static ET maps that do not implement irrigation in the dry season. Hence, the estimated blue ET corresponds to the blue ET amount of past studies in the basin (Msigwa et al., 2021). It is concluded that the representation of seasonal land use dynamics is essential to simulate agricultural (blue and green) water consumption correctly. Also, for land use change studies, it is important to represent the seasonal land use dynamics correctly.

*Data availability.* The openly accessible data used in this analysis are available from the first author upon request (anna.msigwa@nm-aist.ac.tz), the code used in the SWAT+ model is available under the appendices section.

*Author contributions.* AM, CJC and AVG conceived and designed the methodology to simulate evapotranspiration (ET) in SWAT+. AM and CJC Built the models for Kikuletwa catchment using scripts written by CJC. AM performed data analysis and wrote the initial draft of the paper. AVG and HCK supervised the research and contributed to improving the paper prior to submission. AN contributed in data analysis and reviewing the paper at each stage.

*Competing interests.* The authors declare that they have no conflict of interest.

*Acknowledgements:* This work was funded by Flemish Interuniversity Council for University Development Cooperation (VLIR-UOS) through an institutional cooperation (IUC programme) with the Nelson Mandela African Institution of Science and Technology (NM-AIST), under the funded research project "Sustainable Management of Soil and Water for the

Improvement of Livelihoods in the Upper Pangani River Basin", grant number ZIUS2013AP029 and the Academic Open Water Network with grant number JOINT_2019-01-16.

Edited by: Elena Toth

Reviewed by: three anonymous referees

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

## Appendices

### Appendix A. Make Management Script

```python
import sys
     from PIL import Image
     import numpy as np

     def open_tif_as_array(tif_file):
im = Image.open(tif_file)
         imarray = np.array(im)
         return imarray

     def empty_line():
print("")

     def write_to(filename, text_to_write, report = False):
         '''
         a function to write to file
'''
         g = open(filename, 'w')
         try:
             g.write(text_to_write)
             if report:
print('\n\t> file saved to ' + filename)
         except:
             print("\t> error writing to {0}, make sure the file is not open in another program"
     .format(
                 filename))
response = input("\t> continue? (Y/N): ")
             if response == "N" or response == "n":
                 sys.exit()
         g.close

def show_progress(count, end_val, string_before = "percent complete", string_after = "", ba
     r_length = 30):
         percent = float(count) / end_val
         hashes = "#" * int(round(percent * bar_length))
         spaces = '_' * (bar_length - len(hashes))
sys.stdout.write("\r{str_b} [{bar}] {pct}% {str_after}\t\t".format(
             str_b = string_before,
             bar = hashes + spaces,
             pct = '{0:.2f}'.format(percent * 100),
             str_after = string_after))
sys.stdout.flush()
```

```python
    def read_from(filename):
        '''
        a function to read ascii files
        '''
        try:
            g = open(filename, 'r')
        except:
            print("\t> error reading {0}, make sure the file exists".format(filename))
            return
        file_text = g.readlines()
        g.close
        return file_text

class schedule_data:
    def __init__(self, crop_name):
        self.crop_name = crop_name
        self.oct_plant = ""
        self.oct_harvest = ""
        self.aug_plant = ""
        self.aug_harvest = ""
        self.mar_plant = ""
        self.mar_harvest = ""

base_txt = "C:/Users/james/Desktop/root/anna/new/new_swat_plus_model/kikuletwa/Scenarios/De
fault/TxtInOut"
inputs_path = "trajectory_files"

# read trajectory data
trajectories = open_tif_as_array("{base}/{fn}".format(base = inputs_path, fn = "trajectory_
map_thres.tif"))
legend_raw = read_from("{base}/{fn}".format(base = inputs_path, fn = "trajectory_lookup_fin
al.csv"))
dates_raw = read_from("{base}/{fn}".format(base = inputs_path, fn = "crop_plant_harvest.csv
"))

landuse_lum_raw = """landuse.lum: created for trajectories
name                        cal_group           plnt_com                mgt                 cn
2           cons_prac               urban               urb_ro          ov_mann                 tile
                sep                 vfs                 grww                bmp
"""

plant_ini_raw = """plant.ini: created for trajectories
pcom_name           plt_cnt rot_yr_ini  plt_name  lc_status       lai_init        bm_init
 phu_init       plnt_pop        yrs_init        rsd_init
"""

management_raw = """management.sch: created for trajectories
name                        numb_ops   numb_auto                op_typ          mon         day
 hu_sch         op_data1                op_data2        op_data3
"""
```

```python
    landuse_lum = landuse_lum_raw
    plant_ini = plant_ini_raw

trajectories_dictionary = {}
    # trajectory_hru_lum_dict = {}
    crop_schedule_dictionary = {}
    month_dictionary = {'':"None", "Jan": "1", "Feb": "2", "Mar": "3", "Apr": "4", "May": "5",
    "Jun": "6", "Jul": "7", "Aug": "8", "Sep": "9", "Oct": "10", "Nov": "11", "Dec": "12"}
    for line in dates_raw[1:]:
        parts = line.split(",")
        crop_schedule_dictionary[parts[0].lower()] = schedule_data(parts[0])
        crop_schedule_dictionary[parts[0].lower()].oct_plant = "{0}".format(parts[5]).strip("\n
")
        crop_schedule_dictionary[parts[0].lower()].oct_harvest = "{0}".format(parts[6]).strip("
    \n")
        crop_schedule_dictionary[parts[0].lower()].aug_plant = "{0}".format(parts[3]).strip("\n
    ")
crop_schedule_dictionary[parts[0].lower()].aug_harvest = "{0}".format(parts[4]).strip("
    \n")
        crop_schedule_dictionary[parts[0].lower()].mar_plant = "{0}".format(parts[1]).strip("\n
    ")
        crop_schedule_dictionary[parts[0].lower()].mar_harvest = "{0}".format(parts[2]).strip("
\n")

    for line in legend_raw[1:]:
        trajectories_dictionary[line.split(",")[1].lower()] = line.split(",")[2].strip("\n").lo
    wer()
    growing_list = ["FRST", "BANA", "SHRB", "SUGC"]

    for crop_name in trajectories_dictionary:
        # create lum
parts = trajectories_dictionary[crop_name].split("-")
        com_mgt_prefix = "{0}_{1}_{2}".format(parts[0][:3], parts[1][:3], parts[2][:3])
        com_mgt_prefix = com_mgt_prefix.lower()
        if True: #not ((parts[0] == parts[1]) and (parts[0] == parts[2])):
            line_ = "{lum_t}                         null            {plt_comm}  {mgt}          rc_strow_g
cross_slope                  null                null    convtill_nores                  null
                null                null                null                null  \n".format(
                lum_t = trajectories_dictionary[crop_name].lower().replace("-", "_"),
                plt_comm = "{0}_c".format(com_mgt_prefix),
                mgt = "{0}_m".format(com_mgt_prefix),
)
            landuse_lum += line_
            # print(trajectories_dictionary[crop_name])

        # create comm
```

```
comm__ = "{comm_n}_c                    //no           1  \n".format(comm_n = com_mgt_prefix)
           plt_count = 0
           done = []
           for plt in parts:
               if plt == "AGRL":
for agrl_crop in ["TOMA","CORN","SOYB"]:
                       if not agrl_crop.lower() in done:
                           if plt in growing_list:
                               grow_ini = "y"
                           else:
grow_ini = "n"
                           plt_count += 1
                           comm__ += "                                                 {agrl_crop}
         {growing}        0.00000        0.00000        0.00000        0.00000        0.00000     10000
       .00000  \n".format(agrl_crop = agrl_crop.lower(), growing = grow_ini)
done.append(agrl_crop)

                   continue

               if not plt.lower() in done:
if plt in growing_list:
                       grow_ini = "y"
                   else:
                       grow_ini = "n"
                   plt_count += 1
comm__ += "                                                 {plt_l}              {growing
       }        0.00000        0.00000        0.00000        0.00000        0.00000   10000.00000  \n".
       format(plt_l = plt.lower(), growing = grow_ini)
                   done.append(plt)

comm__ = comm__.replace("//no", str(plt_count))
           plant_ini += comm__

           # create_management
           schedule_name = "{0}_m".format(com_mgt_prefix)
number_of_manual_ops = 0
           number_of_auto_ops = 0

           done_2 = []

management_section_head = "{mgt_name}                                   {number_manual}
       {number_auto}   "
           management_section_body = ""
           counter_mgt = 0

for plant_index in range(0, 3):

               date_day_plant = None
               date_mnt_plant = None
```

```python
date_day_harvest = None
             date_mnt_harvest = None
             agrl_list = []

             if plant_index == 0:
agrl_list = ["soyb"]

             if plant_index == 1:
                 agrl_list = ["soyb", "toma"]

if plant_index == 2:
                 agrl_list = ["corn"]

             if parts[plant_index] == "agrl":
                 for agrl_crop_mgt in agrl_list:
if plant_index == 0:
                         date_day_plant = crop_schedule_dictionary[agrl_crop_mgt].mar_plant.spli
     t("-")[0]
                         date_mnt_plant = crop_schedule_dictionary[agrl_crop_mgt].mar_plant.spli
     t("-")[1]
if plant_index == 1:
                         date_day_plant = crop_schedule_dictionary[agrl_crop_mgt].aug_plant.spli
     t("-")[0]
                         date_mnt_plant = crop_schedule_dictionary[agrl_crop_mgt].aug_plant.spli
     t("-")[1]
if plant_index == 2:
                         date_day_plant = crop_schedule_dictionary[agrl_crop_mgt].oct_plant.spli
     t("-")[0]
                         date_mnt_plant = crop_schedule_dictionary[agrl_crop_mgt].oct_plant.spli
     t("-")[1]
                     management_body_line = "
             {activity}{mnt}{day}        0.00000                    {crp}              null       {ord
     er}.00000  ".format(
                         activity = "plnt",
mnt = month_dictionary[date_mnt_plant].strip(" ").rjust(10),
                         day = date_day_plant.rjust(10),
                         crp = agrl_crop_mgt.lower(),
                         order = counter_mgt,
                     )
management_section_body += "{0}\n".format(management_body_line)
                     counter_mgt += 1
                 for agrl_crop_mgt in agrl_list:
                     if plant_index == 0:
                         date_day_harvest = crop_schedule_dictionary[agrl_crop_mgt].mar_harvest.
split("-")[0]
```

```python
                            date_mnt_harvest = crop_schedule_dictionary[agrl_crop_mgt].mar_harvest.
    split("-")[1]
                        if plant_index == 1:
                            date_day_harvest = crop_schedule_dictionary[agrl_crop_mgt].aug_harvest.
    split("-")[0]
                            date_mnt_harvest = crop_schedule_dictionary[agrl_crop_mgt].aug_harvest.
    split("-")[1]
                        if plant_index == 2:
                            date_day_harvest = crop_schedule_dictionary[agrl_crop_mgt].oct_harvest.
    split("-")[0]
                            date_mnt_harvest = crop_schedule_dictionary[agrl_crop_mgt].oct_harvest.
    split("-")[1]

                        management_body_line = "
            {activity}{mnt}{day}         0.00000                     {crp}               null        {ord
    er}.00000  ".format(
                            activity = "hvkl",
                            mnt = month_dictionary[date_mnt_harvest].strip(" ").rjust(10),
                            day = date_day_harvest.rjust(10),
                            crp = agrl_crop_mgt.lower(),
                            order = counter_mgt,
                        )
                        management_section_body += "{0}\n".format(management_body_line)
                        counter_mgt += 1

            elif parts[plant_index] in crop_schedule_dictionary:
                if not parts[plant_index] == "past":

                    if plant_index == 0:
                        date_day_plant = crop_schedule_dictionary[parts[plant_index]].mar_plant
    .split("-")[0]
                        date_mnt_plant = crop_schedule_dictionary[parts[plant_index]].mar_plant
    .split("-")[1]
                    if plant_index == 1:
                        date_day_plant = crop_schedule_dictionary[parts[plant_index]].aug_plant
    .split("-")[0]
                        date_mnt_plant = crop_schedule_dictionary[parts[plant_index]].aug_plant
    .split("-")[1]
                    if plant_index == 2:
                        date_day_plant = crop_schedule_dictionary[parts[plant_index]].oct_plant
    .split("-")[0]
                        date_mnt_plant = crop_schedule_dictionary[parts[plant_index]].oct_plant
    .split("-")[1]

                    management_body_line = "
            {activity}{mnt}{day}         0.00000                     {crp}               null        {ord
    er}.00000  ".format(
                        activity = "plnt",
```

```python
                    mnt = month_dictionary[date_mnt_plant].strip(" ").rjust(10),
                    day = date_day_plant.rjust(10),
                    crp = parts[plant_index].lower(),
                    order = counter_mgt,
                )
                management_section_body += "{0}\n".format(management_body_line)
                counter_mgt += 1

                if plant_index == 0:
                    date_day_harvest = crop_schedule_dictionary[parts[plant_index]].mar_har
vest.split("-")[0]
                    date_mnt_harvest = crop_schedule_dictionary[parts[plant_index]].mar_har
vest.split("-")[1]
                if plant_index == 1:
                    date_day_harvest = crop_schedule_dictionary[parts[plant_index]].aug_har
vest.split("-")[0]
                    date_mnt_harvest = crop_schedule_dictionary[parts[plant_index]].aug_har
vest.split("-")[1]
                if plant_index == 2:
                    date_day_harvest = crop_schedule_dictionary[parts[plant_index]].oct_har
vest.split("-")[0]
                    date_mnt_harvest = crop_schedule_dictionary[parts[plant_index]].oct_har
vest.split("-")[1]

                management_body_line = "
        {activity}{mnt}{day}         0.00000                           {crp}                       null          {ord
er}.00000   ".format(
                    activity = "hvkl",
                    mnt = month_dictionary[date_mnt_harvest].strip(" ").rjust(10),
                    day = date_day_harvest.rjust(10),
                    crp = parts[plant_index].lower(),
                    order = counter_mgt,
                )
                management_section_body += "{0}\n".format(management_body_line)
                counter_mgt += 1

    if counter_mgt == 0:
        continue

    management_raw += management_section_head.format(mgt_name = schedule_name, number_manua
l = counter_mgt, number_auto = number_of_auto_ops) + "\n" +  management_section_body

    # fix hrus based on dictionary

    hru_data_string = """hru-data.hru: for trajectories
id   name                                 topo            hydro            soil
       lu_mgt   soil_plant_init      surf_stor              snow             field
    """
```

```python
hru_data_hru_raw = read_from("{base}/{fn}".format(base = base_txt, fn = "hru-data.hru"))

for line in hru_data_hru_raw[2:]:
    for_part = line
    for i in range(0, 20):
        for_part = for_part.replace("  ", " ")
    parts = for_part.split(" ")
    # print(parts[6].split("_")[0])
    hru_data_string += line.replace(parts[6], trajectories_dictionary[parts[6].split("_")[0
]].lower().replace("-", "_"))

write_to("{base}/{fn}".format(base = 'model_files\Scenarios\Default\TxtInOut', fn = "landus
e.lum"), landuse_lum)
write_to("{base}/{fn}".format(base = 'model_files\Scenarios\Default\TxtInOut', fn = "manage
ment.sch"), management_raw)
write_to("{base}/{fn}".format(base = 'model_files\Scenarios\Default\TxtInOut', fn = "plant.
ini"), plant_ini)
write_to("{base}/{fn}".format(base = 'model_files\Scenarios\Default\TxtInOut', fn = "hru-
data.hru"), hru_data_string)
```

## Appendix B. Trajectories Description

**Table 1B.** Trajectories examples for each fake land use code use for dynamic SWAT+ implementation.

| Map_id | Code | Trajectory |
|---|---|---|
| 1 | TUWO | TUWO-TUWO-TUWO |
| 2 | GRAS | GRAS-GRAS-GRAS |
| 6 | BSVG | BSVG-BSVG-BSVG |
| 11 | FRST | FRST-FRST-FRST |
| 78 | BANA | BANA-BANA-BANA |
| 110 | HMEL | SHRB-SHRB-SHRB |

| 121 | INDN | CORN-BSVG-BSVG |
|---|---|---|
| 146 | LETT | CORN-BSVG-PAST |
| 167 | PAST | PAST-PAST-PAST |
| 182 | SUGC | SUGC-SUGC-SUGC |
| 204 | ASPN | FRST-BSVG-FRST |
| 224 | LIMA | CORN-PAST-PAST |
| 225 | MAPL | CORN-PAST-BSVG |
| 243 | MESQ | CORN-TOMA-PAST |
| 248 | MIGS | CORN-TOMA-BSVG |
| 249 | MINT | TOMA-TOMA-BSVG |
| 254 | MIXC | CORN-TOMA-TOMA |
| 262 | AGRR | AGRL-AGRL-AGRL |

**Table 2B.** Dynamic agricultural land use trajectory and their crop or vegetation cover meaning

| ID | Trajectory | Crop/vegetation cover Meaning |
|---|---|---|
| 1 | CORN-PAST-PAST | rainfed maize-grass-grass |
| 2 | CORN-PAST-BSVG | rainfed maize-grass- sparse vegetation |
| 3 | CORN-TOMA-PAST | rainfed maize- tomato-grass |
| 4 | CORN-TOMA-BSVG | rainfed maize-tomato-sparse vegetation |
| 5 | AGRL-TOMA-BSVG | Beans-tomato-sparse vegetation |
| 6 | CORN-TOMA-IRRM | rainfed maize-tomato-irrigated maize |
| 7 | CORN-PAST-IRRM | Rainfed maize-grass-irrigated maize |

**Table 3B. Land use classes as represented in the Static SWAT+ Model**

| LANDUSE_ID | Land use Class | SWAT_CODE |
|---|---|---|
| 1 | Water | WATR |
| 2 | Grazed grassland | PAST |
| 3 | Grazed shrubland | CRGR |
| 4 | Space vegetation | BSVG |
| 5 | Rainfed Maize | CORN |
| 6 | Irrigated Sugarcane | SUGC |
| 7 | Dense forest | FRST |
| 8 | Sub_Alpine grassland | GRAS |
| 9 | Woodland | TUWO |
| 10 | Mixed Crops | AGRL |
| 11 | Irrigated Banana and Coffee | BANA |
| 12 | Wetland | WEHB |
| 13 | Urban | URMD |
| 14 | Shrubland | SHRB |

## Appendix C. Data used in this study

**Table 1C.** Summary of the different data used in the study with description and sources

| Data Type | Description | Source/ reference |
| --- | --- | --- |
| Climate | Ten station data of rainfall and four stations of maximum/minimum temperature | Tanzania Meteorological Agency (TMA) and Pangani Basin Water Office (PBWO) |
| Digital Elevation Model (DEM) | Elevation data from at 90m resolution | United States Geological Survey (USGS) website |
| Seasonal land use maps | Seasonal land use maps at 30m | (Msigwa et al., 2019) |
| Soil | Africa Soil Information System (AFSIS) at 250m resolution | (Hengl et al., 2015) |
| Remotely sensed based Actual ET | Ensemble ET from six remote sensing products | (IHE Delft, 2020) |
| Land management data | Planting dates, harvesting dates and irrigation application dates and frequency | Farmers interview |