# Peer review of "Representation of seasonal land-use dynamics in SWAT+ for improved assessment of blue and green water consumption"

_Hydrology and Earth System Sciences, 2021_

## Referee Comment (RC3)

[referee-annotated manuscript omitted]

---

## Author Comment (AC2)

**Reply to referee 1**

**Overview**: We would like to thank the reviewer for his/her dedication in reviewing the manuscript. We are also thankful for their considerate and constructive suggestions and comments.

**General Comments:** Overall this paper reports a new approach to the use of an existing hydrological model to better represent African cropping patterns. With water resources (the use and availability of) an important current and future issue for tropical regions, highlighting and documenting a method for improving model outcomes is of use. The paper is well presented, and the methods documented satisfactorily.

**Specific Comments:** Whilst the paper reports the differences between the static and dynamic method in terms of the RMSE and NSE, I would like to have included whether the difference between the two methods results in a statistically significant difference in ET. This would help in showing the magnitude of the difference between the methods. For example, this could be included in the paragraph starting at line 286 where the static, dynamic, and remote sensing methods are compared. Also, line in 371 the authors state "Our study shows a **significant** impact of the representation of seasonal land-use in the SWAT+ model by reducing the errors in water consumption estimations." whereas this has, in fact, not been proven statistically.

**Authors Response:** The point of the reviewer is well taken. During the revision of the manuscript, the statistics results showing whether the ET from the static or dynamic methods will be provided. Also, line 371 in the revised manuscript will be modified based on the statistical results

**Comment:** Were any of the default setting for the land use codes (e.g. PAST) changed in SWAT to better represent African growth? - or are the defaults representative? It would be good to have a sentence relating to this.

**Authors Response:** The setting for static and dynamic models were different. We used the same codes eg PAST as default SWAT+. However, in the dynamic model, the setting for the trajectories didn't mean the default SWAT+ setting. For instance, a placeholder SWAT+ land-use code MIXC signifies trajectory CORN→AGRL→AGRL or MIGS signifies CORN →AGRL →BSVG. This

is explained in line 182 to 183. We will add a detailed explanation to differentiate the two models (Static and dynamic).

**Technical Comments:**

**Comments:** Line 19 (Abstract) The abbreviation for ET has already been defined earlier in the abstract, do not need to do this twice.

**Authors Response:** Line 19, Abbreviation for ET for the second time will be deleted in the abstract.

**Comments:** Line 26 LULC abbreviation is not defined.

**Authors Response:** Line 26, LULC abbreviation will be defined in the revised manuscript.

**Comments:** Line 37 Nitrogen does not need a capital 'N'.

**Authors Response:** Line 37, The comment is well taken. The capital N will be replaced with 'n' in the revised manuscript.

**Comments:** Line 38 LAI abbreviation is not defined (unless I missed it).

**Authors Response:** The LAI abbreviation will be defined in the revised manuscript.

---

## Author Comment (AC3)

**Reply to referee 2**

**Overview**:  We would like to thank the reviewer for his/her dedication in reviewing the manuscript. We are also thankful for his/her detailed and constructive suggestions and comments. We have addressed all the comments raised by the reviewer and the manuscript has improved from the proposed changes.

The authors evaluate a method to depict seasonal land use dynamics with SWAT+. Moreover, they evaluate blue and green ET for the study area. The results with regard to the implementation of seasonal land use dynamics evaluate by using satellite ET are promising. However, more details on the model and the implementation need to be provided, before this manuscript can be considered for publication.

**General comments:**

1) There are two topics in the manuscript that are not very well related. E.g. the State-of-the art focuses on the implementation of seasonal land use dynamics. However, also blue and green ET is also one of the study aims and not well represented in the introduction section. Most of the paper is about seasonal land use dynamics. The manuscript part on blue and green water consumption is not very well connected to this. In parts, it reads like a different paper. Particularly as in the last part of the results section a new method is presented that was not introduced in the methods section. I would suggest that the authors either focus on the topic of seasonal land use change implementation and its impacts (which might include blue and green ET as one -but not the only- example), or they provide more motivation why blue and green ET is important in this context and why these two topics should be dealt with in one manuscript. In this case, please also include blue and green water in the state-of-the-art.

**Authors Response:** We agree with the reviewer, after re-reading our previous manuscript, we realize we have two topics as stated by the reviewer and less details are provided in the introduction on blue and green ET. We will focus on blue and green ET, the state of art will be included and more details on the introduction will be included in the revised manuscript.

Our innovation is that we use an agro-hydrological model (SWAT+) to represent blue and green ET for different cropping seasons (represented by trajectory with time and space) and we further used validated remote sensing ET to evaluate the simulated ET from SWAT+.

2) The model calibration and validation approach are not clear. Details need to be provided to judge on the validity of the results.

**Authors Response:** We did not calibrate the SWAT+ model, our aim was to improve the spatial distribution of blue and green water consumption and not discharge simulation. We evaluated the simulated ET by comparing with the remote sensing ET. We will include a clear explanation in the revised manuscript under the methodology section.

3) Model setup for static and dynamic model needs to be explained in detail. Inconsistencies in model outputs, e.g. static does not equal dynamic ET for areas that are static in both model implementations, should be explained.

**Authors Response:** We agree with the reviewer, the revised manuscript will include more explanation of the model setup under the methodology section.

4) Land use data (e.g. land use classes, trajectories, accuracies) need to be shown in more detail.

**Authors Response:** We will provide enough details on the land use classes, trajectories and accuracies in the revised manuscript.

5) Innovative aspects of your research should be highlighted and presented against the state-of-the-art.

**Authors Response:** We agree with the reviewer that the state of art of the manuscript is not clear, we will include more explanation to our specific innovation. For example we will explain in detail our innovation as the use of agro-hydrological model such as SWAT+ to representing blue and green ET for different cropping seasons (represented by trajectory with time and space) and the use of remote sensing ET to evaluate the simulated ET from SWAT+.

6) Proof-reading by a native speaker would be helpful. I suggested some changes, but there are certainly more sentences that need to be improved.

**Authors Response:** Reviewer's comment is taken; we will have a native English speaker to help read and edit the revised manuscript.

**Line specific comments:**

l.9-10: Please clarify and unify terms: cropping cycle, cropping seasons

**Authors Response:** Cropping cycle and cropping seasons were used interchangeably. We will stick with the cropping seasons instead of cropping cycle.

l.11: 'In most agro-hydrological model applications such as SWAT+ in Africa, only one cropping season per year is represented.' This is indeed surprising. Please see also my comment on l. 56 and l. 72-73.

**Authors Response:**

l.14: Better focus on the topic of this paper in the abstract: 'This study builds upon earlier research that proposed an approach on how to incorporate seasonal land use dynamics in the SWAT+ model but mainly focused on the temporal pattern of LAI and tested the approach in a small catchment (240 km2).'

**Authors Response**

l.20: suggest to change to: , remote sensing estimates, resulting in a higher performance' remove ‚than default'

**Authors Response:** Thank you the comment is taken we will revise and change in the revised manuscript.

l.22-23: Please improve the language and strengthen conclusion

**Authors Response:** We agree with the reviewer, the conclusion will be improved.

l.30 suggest ‚at the' instead of ‚per'

**Authors Response:** We will change from "per" to "at the"

l.36: I believe these are studies that have implemented land-use dynamics. In this case, ‚few' is misleading, suggest to say ‚A few…'

**Authors Response:** Comment is well taken. We will change from "Few" to "A few"

l.44-45: Please clarify, what you mean with ‚implemented seasonal land-use dynamic in SWAT and SWAT+ through land-use trajectories, and not land-cover classes." As I understand it, a trajectory is also a change of land-use and land-cover classes. So that the meaning of the sentence is not clear to me.

**Authors Response:** After re- reading the sentence yes, it is misleading. We will revise and provide a better explanation in the revised manuscript. We mean to say that, **a** study by Nkwasa et al. 2020 and our study, use the land use trajectories as input map that shows changes in land use per time and space, unlike other studies where the input map is a land use map then the seasonal changes in the land use is implemented later by crop rotation.

l.56: AfricaN basins

**Authors Response:** Africa will be changed to African

l.56 and 72-73:‚…typically not represent different cropping seasons' and 'Although the SWAT (+) model is capable of representing multiple cropping seasons, this is rarely implemented.'

I agree with you, that it is important to represent different cropping seasons. But please reflect that seasonal crop rotations can be depicted with SWAT and that has been done in the past in study areas with a strong seasonality, e.g. typically in India. Please find 3 example studies below. For these implementations the seasonal changes within one year is however always the same. Would it be possible to go beyond that with your methodology? Do you account for all possible combinations of seasonal crop rotations in space? Please highlight the innovation in your research.

Garg, K.K., Bharati, L., Gaur, A., George, B., Acharya, S., Jella, K. and Narasimhan, B. (2012), Spatial mapping of agricultural water productivity using the swat model in the Upper Bhima catchment, India. Irrig. and Drain., 61: 60-79. https://doi.org/10.1002/ird.618

Narsimlu, B., Gosain, A.K. & Chahar, B.R. Assessment of Future Climate Change Impacts on Water Resources of Upper Sind River Basin, India Using SWAT Model. Water Resour Manage 27, 3647–3662 (2013). https://doi.org/10.1007/s11269-013-0371-7

Wagner, P. D., Kumar, S., and Schneider, K.: An assessment of land use change impacts on the water resources of the Mula and Mutha Rivers catchment upstream of Pune, India, Hydrol. Earth Syst. Sci., 17, 2233–2246, https://doi.org/10.5194/hess-17-2233-2013, 2013.

**Authors Response:** Thank you for your comment. yes we can go beyond these studies with our innovation since we are representing the trajectories which include changes in time and space. We see an example of study in Pangani by Ndomba et al. 2008, the cropping seasons were not implemented because land use classification in the tropics is challenging as indicated by unclearly defined land use practices in this study area (Ndomba et al., 2008).

The paper by Merriman et al. (2019) is another example where the crop rotation has been implemented in details and where our innovation could be adopted. The study has shown crop rotation in detail e.g. "one crop cover, changing from corn silage to cereal rye" however, this method might not apply in most of the tropical African catchments like our catchment, because one crop cover may have different crop rotation practices. Hence, we need to know the location (having a specific rotation). For example, we have indicated that the same rainfed maize crop cover may change in one field and form a trajectory CORN→AGRL→AGRL or change from CORN →AGRL →BSVG. Thus, it is very crucial to use the changes in seasonal land use to represent these changes in space and time.

l.74: 'By default, SWAT simulates a single growing cycle every year.' This is true, but it can be argued that the modeler should adjust the default, if the default is not applicable.

**Authors Response:** We agree with the reviewer's argument. We will revise the sentence and add more literature on how most of the modelers have and have not adjusted the default model to highlight our innovation.

l.80-83: Please outline stronger what the new contribution of this paper is. If it building on earlier findings is fine, but this could also be outlined in the methods section.

**Authors Response:** Thank you for your comment. This is well noted. Our innovation is that we use an agro-hydrological model (SWAT+) to represent blue and green ET for different cropping seasons (represented by trajectory with time and space). Additionally, we used validated remote sensing ET to evaluate the simulated ET from SWAT.

We will restructure the introduction and focus more on the blue and green ET. We will include the detail explanation of our innovation in the revised manuscript.

l.80-92: Suggest to shorten the paragraph to the aims. Please move the methodological details to the methods section.

**Authors Response:** The comment is well taken. We will move the methodological details of the last paragraph in the introduction to the methods section.

l.95 As there has been SWAT research on the Pangani basin, I would suggest to relate your research (literature review + findings) to it. See e.g.:

Notter, B., Hurni, H., Wiesmann, U., and Abbaspour, K. C.: Modelling water provision as an ecosystem service in a large East African river basin, Hydrol. Earth Syst. Sci., 16, 69–86, https://doi.org/10.5194/hess-16-69-2012, 2012.

**Authors Response:** The comment is well taken. We acknowledge related works using SWAT in the Pangani basin. Hence, we will relate their research findings to our study in the revised manuscript.

Fig. 1: Inset map is not readable. Please revise.

**Authors Response:** The revised version of the manuscript will include a revised figure 1.

Fig. 2: It would be preferable to show a 30 year average of rainfall to depict the climate, if data is available. The authors state that there was at least data available for 2006-2013, l.122. Certainly, a longer period would be better. This figure is also depicted in Msigwa et al. 2019. Please, make sure that there are no copyright issues. You may include temperature to provide a bit more information here.

**Authors Response:** We agree with the reviewer. The average of 30 years will be analysed and included in the revised manuscript. Also, we will include temperature averages to provide more information.

l.115: Please add which DEM was used, not only the source for download. SRTM?

**Authors Response:** The source of DEM will be added to the revised manuscript. DEM is SRTM from NASA.

l.123-126: As the entire paper relies on the accuracy of these land use maps, you need to provide classification accuracies here. I would suggest to show at least overall accuracy and the range of user accuracies for the different land use classes. Please also state which and how many classes have been identified and which classification algorithm was applied.

**Authors Response:** The information on the classes accuracies will be included in the revised manuscript.

l.127: 'For instance,…' One example is not sufficient. Either provide the setup information for all land use classes or refer the reader to a publication where you have shown that.

**Authors Response:** The comment is well noted. We will refer the reader to the additional setup information in the appendix within the revised manuscript.

l.136: Full stop missing

**Authors Response.** Thank you, we will include the fool stop.

l.145-147: Sentence and reasoning not clear to me. Bananas and coffee should probably not change within a year. Did they in the trajectory analysis? If so, how would you explain that? Also, how would you parameterize a combined class of coffee and bananas? Please clarify.

**Authors Response:** We didn't implement the trajectory analysis for banana and coffee. However, in the catchment we have farms that change from banana and coffee to banana, coffee and maize because they plant maize during the rainy season. This analysis was explained in the previous paper by Msigwa et al. 2019.

Figure 3: While this map provides a good first overview, regarding the topic of the paper, I think it is necessary to show the different land use trajectories in more detail.

**Authors Response:** The comment is well taken, we will revise the map and provide different land use trajectories in more detail.

l.164: Otherwise spelled as 'sub-basins', please unify.

**Authors Response:** Sub Basin will be changed to sub-basin in the whole revised manuscript.

l.170: Are you using the option to grow two or more crops at the same time? If yes, this should be highlighted, if not, why mention this?

**Authors Response:** We did not grow one crop at the same time. We will remove the highlight as suggested by the reviewer.

l.175-176: suggest to revise to 'limited amount of input data'

**Authors Response:** Reviewer comment is taken. We will rephrase in the statement in the revised manuscript.

l.177: 'rather than using remote sensing climate data' Sentence not clear, please clarify.

**Authors Response:** Reviewer comment is taken we will rephrase the statement in the revised manuscript.

l.181: Table 1B+2B do not show 40 trajectories, please clarify. Also, some of the trajectories seem to be no real rotations, e.g. "indn CORN-BSVG-BSVG", seems to be a single crop corn in one cropping season and no cropping in the other seasons. I think it should be highlighted which of these trajectories describe real crop rotations and which are only single crops, which could probably be well represented by a model without a seasonal representation of crops.

**Authors Response:** The comment is well taken. We will revise and include all the 40 trajectories and specify the trajectories with only the cropping seasons.

Table 1: I would suggest to write 2-3 sentences to explain the shown management file highlighting the capabilities, e.g. tomato and soy bean are grown on the same field. Suggest to delete white space. Moreover, if you have tomato and soy bean on one field, how was that derived in the land use classification? And if this was a class for itself, how good was the classification performance?

**Authors Response:** The reviewer comment is well taken. We will include a detailed explanation of the management file with the implementations. The tomato and soy bean were grown on one field at different times, the land use class was term as "irrigated mixed crop".

l.217: I cannot find the source , IHE Delft, 2020' in the reference section.

**Authors Response:** The references will be reviewed and IHE Delft 2020 will be included in a revised manuscript.

l.239: ,statistical matrices'?

**Authors Response:** Thank you for the comment, we will revise and change to model evaluation statistics as suggested by Moriasi et al. (2007).

l.213-239 The Model Evaluation section needs a thorough revision, please address the following points:

**Authors Response:** We will revise the section and add all the details suggested by the reviewer in the revised manuscript.

1) Setup of the two models: Which land use map was used for the static model?

**Authors Response:** The March land use was used for static model. This will be clearly stated in the revised manuscript.

2) Calibration approach? Did you calibrate your models? How did you do that and did you do this separately for the static and dynamic model?

**Authors Response:** We did not calibrate any model static or dynamic model. Our aim was to improve the spatial distribution of blue and green water consumption and not discharge simulation. We compared the static and dynamic model in default conditions. This approach allowed us to compare model results in default parameter conditions, considering parameter calibrations vary with different catchments. Nkwasa et al., (2020) also suggested that improved representation of crop and agricultural land use processes should precede any model calibration efforts. Thus, we evaluated the simulated ET by comparing with the remote sensing ET. This will be clearly stated in the revised manuscript.

3) It seems as if the model performance is solely evaluated with ET. This needs a better justification and explanation. What about the discharge data described in the methods section? Please provide more information on the ET data used for calibration (?) and validation. What exactly was compared? Basin values, sub-basin values, grid values? If that has been carried out in a previous study, you may also refer to that study for details, but you need to provide the reader with the main information that is necessary to evaluate the performance of your model.

**Authors Response:** The reviewer comment is well taken. The model performance was evaluated by the ET only. Hence, it was unnecessary to include the discharge data that was not used in the description. This will be omitted. Also, the description of how we compared and evaluated the remote sensing ET with the model simulated ET will be added in the revised manuscript.

See also the following HESS paper on SWAT modeling with ET data in Africa:

Odusanya, A. E., Mehdi, B., Schürz, C., Oke, A. O., Awokola, O. S., Awomeso, J. A., Adejuwon, J. O., and Schulz, K.: Multi-site calibration and validation of SWAT with satellite-based evapotranspiration in a data-sparse catchment in southwestern Nigeria, Hydrol. Earth Syst. Sci., 23, 1113–1144, https://doi.org/10.5194/hess-23-1113-2019, 2019.

4) Actually the indices that were applied are well known. I would suggest to rather focus on explaining the calibration and validation strategy and do not explain the indices in such detail.

**Authors Response:** The comment is well taken. An explanation on the indices will be omitted in the revised manuscript and more information on how we evaluated the ET will be added.

5) For which period was the model run?

**Authors Response:** The model run from a period of 2008 to 2013. The year 2006 and 2007 was a warming period. Details will be added in the revised manuscript.

l.253: Verb missing

**Authors Response:** The sentence will be revised and a verb will be added in the revised manuscript.

l.260-262: Please explain and clarify, sentence not clear to me.

**Authors Response:** The sentence will be paraphrased and made clear in the revised manuscript.

l.266-268: Please revise sentence and check grammar.

**Authors Response:** The comment is well taken and we will revise the sentence.

Fig. 4: How come that the static ET peaks are some times higher than the dynamic ones? I would have assumed that dynamic ET =static ET for the period in which both have the same crop and that for all other seasons dynamic ET > static ET. As detailed and required information on how the static land use was implemented (and differs from the dynamic land use) is missing (see previous comment), it is hard to understand these differences.

**Authors Response:** We agree with the reviewer comment, detailed explanation of the model setup is needed to know the differences. The model setup for static used a March land use map with only 14 land use classes, while the dynamic model used a land use map with 40 trajectories. Hence, the changes in the ET might be due to the different land use maps yielding different number of HRUs. A clear explanation will be added in the revised manuscript.

l.279: Suggest 'A notable difference…'

**Authors Response:** Comment is taken. We will change "The" to "A"

l.281: Please define what you refer to as 'mass balance in percentage'

**Authors Response:** Mass balance was meant to say change in soil water balance in the model. In the revised manuscript we will modify to "change in soil water balance".

l.286-292 and Fig.5: How do you explain the strong differences for the areas that show a high satellite ET? Even the dynamic model underestimates these considerably.

**Authors Response:** The ET from dynamic could not reach maximum satellite ET because the satellite ET estimates also have uncertainties in the mountainous areas because of the presence of cloud cover. There are no observation data in these areas that we can validate with.

l.293-294: It is hard to follow the line of argumentation here. Looking at Figure 5 I see most pronounced changes between static and dynamic implementation at the Northern border of the catchment. But when I look at Fig. 2, these are not areas with trajectories. Please explain these differences. I would expect that all areas with no trajectories show the same ET value in both models.

**Authors Response:** Thank you for the comment. We agree with the reviewer's argument. We acknowledge that we have not explained how we corrected some unrealistic trajectories in the land use maps when implementing trajectories in the dynamic model. When implementing the trajectories, some unrealistic trajectories were noticeable for example; a trajectory of irrigated banana and coffee land use to forest land use to forest land use in the March, August and October maps respectively is unrealistic Therefore, we change that trajectory to be forest land use in the

dynamic model which could be the reason why the static and dynamic ET could be different even for the static land use map in some regions. This information will be clearly elaborated in the revised manuscript.

l.295-297: Please clarify the following sentence: "Likewise, the changes seen in the high land areas of irrigated banana and coffee and the forested areas might be due to the increase in the number of HRUs in the dynamic SWAT+ model that contributed to the more accurate results." Why do HRU numbers change? Again the implementation differences between static and dynamic scenario are not clear. From a methodological point of view, I would not expect changes in the number of HRUs. For your study aims you need to make sure that you minimize any other impact (e.g. differences in model structures) to really deduce the impact of your seasonal land use change implementation.

**Authors Response:** The reviewers' comment is well taken. We will add more details on the implementation of static and then dynamic models. The number of HRUs will be different because the input land use maps have different number of land uses classes. In the dynamic model the land use map (trajectory map) had 40 land use classes while the static land use map had 14 land use classes. However, we maintain the same number of sub-basins to try and minimize the differences in model structures.

l.308-309: Please improve language ,for annual (Figure 6) and from 2008 to 2013.'

**Authors Response:** The caption language will be improved for figure 6.

l.320: As mentioned earlier: Please include a land use (trajectory) map, I cannot see where sugarcane is located. The reader must be able to follow and verify your conclusions.

**Authors Response:** The point is well taken. We will revise the land use map and include the trajectories.

l.324-335: These methods have not been explained. If you want to show these here, you need to include them in the methods section. It also looks as if some data from a forthcoming publication

is shown. Please specify if you refer to the data or to the methods with the reference. See also my general comment on the two topics covered in this manuscript.

**Authors Response:** The comment is well taken. We will include an explanation of the methods for estimating blue and green ET that we have compared with the SWAT. We will also specify that we have used the data from a previous paper (Msigwa et al., 2021) on the revised manuscript.

Msigwa, A., Komakech, H. C., Salvadore, E., Seyoum, S., Mul, M. L., & van Griensven, A. (2021). Comparison of blue and green water fluxes for different land use classes in a semi-arid cultivated catchment using remote sensing. *Journal of Hydrology: Regional Studies*, *36*, 100860.

l.335: Forthcoming Msigwa et al. 2020 paper is not available in the reference section

**Authors Response:** Thank you for the comment we will include the reference in the reference section of the revised manuscript.

l.340: Please be more careful with this statement ,none of these studies represented seasonal dynamics'. As outlined above, there are a number of studies that have incorporated seasonal crop rotations in India and possibly also elsewhere. They might not have compared the effect to a static model, but they still implemented them. Please highlight what the advantage of your approach is. One example might be the spatial representation of trajectories.

**Authors Response:** The comment is well taken we will revise the statement and add that our approach goes further to represent the spatial location of these seasonal changes.

l.342: Typo: You did show that, didn't you?

**Authors Response:** We meant the study by Nkwasa et al. 2020 didn't show how the seasonal land-use dynamic improved water balance component such as ET but not our current study. We will revise the sentence to make it clear in the revised manuscript.

l.349-352: Please also discuss and explain, why static and dynamic ET do not match for static land use areas and why your ET estimate never reaches the maximum satellite ET.

**Authors Response:** The comment is well taken. We will include the explanation in detail in the revised manuscript.

ET from static and dynamic model could not much in most areas due to the correction of the land use trajectories as explained earlier. However, this is a very important observation we will run the models again to have to confirm our argument.

The ET from dynamic could not reach maximum satellite ET because the satellite ET estimates also have uncertainties in the mountainous areas because of the presence of cloud cover. There are no observation data in these areas that we can validate with (Msigwa et al., 2021).

l.355: Forthcoming Msigwa et al. 2021 paper is not available in the reference section

**Authors Response:** Thank you for the comment. We will include the reference in the reference section of the revised manuscript.

l.365: What about the uncertainties of the land use maps and the associated trajectories as well as their impact on hydrology? Mostly it is hard to assess land use with multi-spectral satellite data in all seasons due to cloud cover (in the rainy season). How did you deal with this? And what does this mean for the transferability of your methodology?

**Authors Response:** We agree with the reviewer's comment. There were uncertainty associated with the trajectories for example unrealistic trajectories like change from crop to forest then crop again. These types of trajectories were corrected and reclassified. We will include these details in the discussion section.

We only used the images with less than 10% of the cloud cover. We also mask the clouds by replacing with the proper land use either using our field survey information or previous land use map by Kiptala et al. (2013). This approach had some effects since we were not able to capture the peak wet seasons where the crops have fully grown. We will include this information in the discussion section of the revised manuscript.

l.338: Please include a discussion of model performance in the discussion section.

**Authors Response:** The reviewer's comment is well taken. The revised manuscript will include model performance in the discussion section.

l.385: ‚blue water amount is in line with previous studies' Not sure to which section the authors refer here and to which studies. Please clarify.

**Authors Response:** The reviewer's comment is well taken. It is true that the sentence needs a reference as to which study, we mean. We will provide clarification in the revised manuscript.

---

## Author Comment (AC4)

**Reply to Referee 3**

**Overview**: We would like to thank the reviewer for his/her dedication in reviewing the manuscript. We are also thankful for his/ her consideration and constructive suggestions and comments.

**Reviewer summary**: The paper "Representation of seasonal land-use dynamics in SWAT+ for improved assessment of blue and green water consumption" reports an application of the SWAT+ model in Africa. The authors implemented seasonal dynamic land-use in SWAT+ in order to improve vegetation growth simulation and to obtain more realistic temporal patterns of the blue and green water consumption from simulated evapotranspiration. Results of the simulations (static and dynamic seasonal land use) in terms of ET were compared to the ET values estimated by using remote sensing. The authors concluded that the seasonal land-use dynamic approach produces better ET results, which provides better estimations of blue and green water.

General comments:

**Reviewer comment**: The paper is very similar to a previous work that has been published in 2020 by the same research group (Nkwasa et al., 2020). The latter paper showed a better performance of the SWAT+ model by using the seasonal land use dynamic (ET at HRU level after the implementation of trajectories in SWAT+ model was compared to the default SWAT+ model). In addition, in 2019 the authors published a study carried out in the same basin applying SWAT+ with dynamic land use and the authors concluded that detailed seasonal land use maps are essential for quantifying annual irrigation water use of catchment areas. For these reasons, it seems difficult to find the novelty of the present paper. Hence, I invite the author to revise the introduction in order to better focus on the advancement of knowledge proposed in this study. Taking into account that the methodological approach (seasonal land-use dynamics in SWAT+) has already been published, the authors should better focus on the green and blue waters.

I suggest major revisions, the current version cannot be published in HESS.

**Authors Response**: The reviewer comment is well taken. The manuscript will wholly be revised to focus on the blue and green water.

To make a clear distinction with earlier studies, we will first introduce blue and green water, specifically how it is important but difficult to map. However, mapping the blue and green water is possible with agro-hydrological models such as SWAT but then they need a better representation of the seasonality/cropping seasons. Hence, this can be done by the trajectory approach in SWAT+ as suggested by Nkwasa et al. (2020). The innovation is that we use an agro-hydrological model (SWAT+) to represent blue and green water for different cropping seasons. Additionally, we use remote sensing ET for evaluation of simulated model ET.

Methodology:

**Reviewer comment**: a better description of the remote sensing ET evaluation is needed, the reference IHE Delft, 2020 is not listed.

**Authors Response:** The reviewer comment is well taken. We will give details on the remote sensing ET and also include IHE Delft; 2020 in the reference list.

**Reviewer comment**: More details on irrigation are needed, for instance, the source of water for irrigation (i.e. from the river, shallow aquifer, etc). Analyzing table 1, it seems that the option auto-irrigation was used. Please explain it. Did the author compare the amount of auto-irrigation to the actual irrigation (data provided by farmers)?

**Authors Response:** We agree with the reviewer suggestion. The revised manuscript will include the details of irrigation such as the source of water (river). We will also evaluate the auto- irrigation in comparison with the actual irrigation through the information gathered in the field. All the details will be added in the revised manuscript.

**Reviewer comment**: In my opinion, the equations and description of the RMSE, PBIAS, and NSE are not necessary.

**Authors Response:** We agree with the reviewer comment. We will not include the description of the RMSE, PBIAS and NSE in the revised manuscript.

**Reviewer comment**: For which period was the model run?

**Authors Response:** The model run from a period of 2008 to 2013. The year 2006 and 2007 was a warming period. Details will be added in the revised manuscript.

**Reviewer comment**: Figure 2 has already been reported in Msigwa et al., 2019 and for this reason, I suggest do not report it here.

**Authors Response:** We agree with the comment. However, we plan to revised the figure and add new information like temperature and take a long analysis of rainfall like 30 years. According to reviewer 2 comments.

**Reviewer comment**: I suggest adding a new map in figure 3 with the land use (static land use).

**Authors Response:** The comment is well taken. We will add information on both the trajectory and static land use maps.

**Reviewer comment**: Calibration needs a better presentation. It seems that the calibration was performed for the static and dynamic approach, please show the calibrated parameters in both simulations. A table with calibrated parameters for both simulations is expected. What about validation?

**Authors Response:** We did not calibrate the SWAT+ model, our aim was mainly to improve the spatial distribution of blue and green water consumption and not discharge simulation. We evaluated the simulated ET by comparing with the remote sensing ET. We will include more explanation in the revised manuscript's methodology section.

Result section:

**Reviewer comment**: Methods reported in Lines from 324 to 335 are not reported in the "Material and methods" section. What is the aim to show them in Figure 8? In my opinion, this section should be eliminated.

**Authors Response:** The comment is well taken. We will include a brief explanation of the methods for estimating blue and green ET that we have compared with the SWAT+ model results in the revised manuscript. The aim of comparing blue and green ET estimates was to provide evidence that the blue ET estimates from dynamic SWAT+ model show no statistical difference with the blue ET estimated using remote sensing with Van Eekelen method.

**Reviewer comment**: Line 335. Caption Figure 8. Msigwa et al. 2020 is not reported in the references. Is this reference the same as that reported in Line 355 Msigwa et al. 2021 (missed in the reference)?

**Authors Response:** Thank you for the comment we will include the references in the reference section in the revised manuscript.

Discussions:

**Reviewer comment**: Innovative aspects of your research should be highlighted and presented against the state-of-the-art. The authors reported (LINE 355) that blue and green ET estimates from SWAT+ for the mixed crop land-use show no significant difference in the values from the two methods (EK and SWB) assessed in the upcoming paper by Msigwa et al., (2021). This is not the aim of the present paper.

**Authors Response:** The comment is well taken and Line 355 will revised. We will present our research results against the state-of-the-art in the revised manuscript.

**Reviewer comment**: Please discuss the difference between Figure 5b and fig. 5c and their comparison with figure 7. I did not understand why static and dynamic ET do not match for static land use areas. In the upper right corner, figure5b shows the green areas in correspondence with the static land use (see fig 3). The large difference is difficult to explain with a different number of HRUs. In addition, a large difference remains between dynamic and satellite ET (Fig 5a and 5b) that needs to be explained.

**Authors Response:** The reviewer's comment is well taken. Detail explanation of the relationship between figure 5a, 5b and figure 7 will be provided in the revised manuscript.

The number of HRUs will be different because the input land use maps have different number of land uses classes. Dynamic model the land use map (trajectory map) had 40 land use classes while the static land use map had 14 land use classes. However, the same number of subbasins is maintained to try and preserve a similar model structure.

We acknowledge that we have not explained how we corrected some unrealistic trajectories in the land use maps when implementing trajectories in the dynamic model. When implementing the trajectories, some unrealistic trajectories were noticeable for example; a trajectory of irrigated banana and coffee land use to forest land use to forest land use in the March, August and October maps respectively is unrealistic Therefore, we change that trajectory to be forest land use in the dynamic model which could be the reason why the static and dynamic ET could be different even for the static land use map in some regions. This information will be clearly elaborated in the revised manuscript.

The ET from dynamic could not reach maximum satellite ET because the satellite ET estimates also have uncertainties in the mountainous areas because of the presence of cloud cover. There are no observation data in these areas that we can validate with.

**Reviewer comment**: Please discuss differences in water balance components between static and dynamic scenarios.

**Authors Response:** The reviewer's comment is well taken. Explanation of differences in water balance components between static and dynamic will be included in the revised manuscript.

**Reviewer comment**: Please discuss the limit of the present study.

**Authors Response:** The point is taken. We will include the limitation of our study in the discussion section of the revised manuscript.

**Reviewer comment**: Conclusions need to be improved.

**Authors Response:** We agree with the reviewer's comment. We will improve the conclusion section in the revised manuscript.

**Reviewer comment**: The authors reported, "The maps with calculated blue water use from the dynamic SWAT+ model correspond to the known irrigated area and the calculated blue water amount is in line with previous studies". The first assertion is obvious since the authors set the irrigation in that areas. The second was expected since the authors refer to their previous papers.

**Authors Response:** We agree the statements made are obvious considering we have implemented the irrigation in the specific areas. We will clearly rephrase the sentence to mean that with the dynamic model, irrigation was implemented especially during the dry season unlike in the static model where there was no irrigation implementation. Thus, the static model does not give reliable estimates of blue ET.

**Reviewer comment**: The paper needs to be carefully checked for typing errors (see some of them have been highlighted in the file enclosed)

**Authors Response:** The reviewers typing error corrections are noted. We will implement all the corrections and check for other typing errors in the revised manuscript.

---

## Author Comment (AC5)

**Reply to referee 1**

**Overview**:  We would like to thank the reviewer for his/her dedication in reviewing the manuscript. We are also thankful for their considerate and constructive suggestions and comments.

Note: The lines are according to simple markup

**General Comments:** Overall this paper reports a new approach to the use of an existing hydrological model to better represent African cropping patterns. With water resources (the use and availability of) an important current and future issue for tropical regions, highlighting and documenting a method for improving model outcomes is of use. The paper is well presented, and the methods documented satisfactorily.

**Specific Comments:** Whilst the paper reports the differences between the static and dynamic method in terms of the RMSE and NSE, I would like to have included whether the difference between the two methods results in a statistically significant difference in ET. This would help in showing the magnitude of the difference between the methods. For example, this could be included in the paragraph starting at line 286 where the static, dynamic, and remote sensing methods are compared. Also, line in 371 the authors state "Our study shows a **significant** impact of the representation of seasonal land-use in the SWAT+ model by reducing the errors in water consumption estimations." whereas this has, in fact, not been proven statistically.

**Authors Response:** We have included the statistics results showing that there is significant difference between the ET from static model and that of the dynamic model. Line 310- 312 was added.

**Comment:** Were any of the default setting for the land use codes (e.g. PAST) changed in SWAT to better represent African growth? - or are the defaults representative? It would be good to have a sentence relating to this.

**Authors Response:** We did not change the default parameters of the land use codes except the maximum potential leaf area index (BLAI) for maize. This was adjusted based on the field

measured data. A sentence relating to this has been added in the revised manuscript under section 2.6. Line 211-213

**Technical Comments:**

**Comments:** Line 19 (Abstract) The abbreviation for ET has already been defined earlier in the abstract, do not need to do this twice.

**Authors Response:** The abstract was edited hence the ET has not been defined twice.

**Comments:** Line 26 LULC abbreviation is not defined.

**Authors Response:** Both abstract and introduction have been changed. We careful considered the comment in line 78.

**Comments:** Line 37 Nitrogen does not need a capital 'N'.

**Authors Response:** Line 66, The capital N has been replaced with 'n' in the revised manuscript.

**Comments:** Line 38 LAI abbreviation is not defined (unless I missed it).

**Authors Response:** The LAI abbreviation has been defined in the revised manuscript (line 67).

---

## Author Comment (AC6)

**Reply to referee 2**

**Overview**: We would like to thank the reviewer for his/her dedication in reviewing the manuscript. We are also thankful for his/her detailed and constructive suggestions and comments. We have addressed all the comments raised by the reviewer and the manuscript has improved from the proposed changes.

Note: All the referred lines are in simple markup.

The authors evaluate a method to depict seasonal land use dynamics with SWAT+. Moreover, they evaluate blue and green ET for the study area. The results with regard to the implementation of seasonal land use dynamics evaluate by using satellite ET are promising. However, more details on the model and the implementation need to be provided, before this manuscript can be considered for publication.

**General comments:**

1) There are two topics in the manuscript that are not very well related. E.g. the State-of-the art focuses on the implementation of seasonal land use dynamics. However, also blue and green ET is also one of the study aims and not well represented in the introduction section. Most of the paper is about seasonal land use dynamics. The manuscript part on blue and green water consumption is not very well connected to this. In parts, it reads like a different paper. Particularly as in the last part of the results section a new method is presented that was not introduced in the methods section. I would suggest that the authors either focus on the topic of seasonal land use change implementation and its impacts (which might include blue and green ET as one -but not the only- example), or they provide more motivation why blue and green ET is important in this context and why these two topics should be dealt with in one manuscript. In this case, please also include blue and green water in the state-of-the-art.

**Authors Response:** We agree with the reviewer, after re-reading our previous manuscript, we realize we have two topics as stated by the reviewer and less details are provided in the introduction on blue and green ET. We have focused on blue and green ET, the state of art is included and the whole introduction has been restructured in the revised manuscript.

Our innovation is that we use an agro-hydrological model (SWAT+) to represent blue and green ET for different cropping seasons (represented by trajectory with time and space) and we further used validated remote sensing ET to evaluate the simulated ET from SWAT+.

2) The model calibration and validation approach are not clear. Details need to be provided to judge on the validity of the results.

**Authors Response:** We did not calibrate the SWAT+ model, our aim was to improve the spatial distribution of blue and green water consumption and not a discharge simulation. We evaluated the simulated ET by comparing with the remote sensing ET. A detailed explanation has been added in the model evaluation section.

3) Model setup for static and dynamic model needs to be explained in detail. Inconsistencies in model outputs, e.g. static does not equal dynamic ET for areas that are static in both model implementations, should be explained.

**Authors Response:** We agree with the reviewer, we have added details about model configuration and implementation of trajectories in sections 2.5 and 2.6.

4) Land use data (e.g. land use classes, trajectories, accuracies) need to be shown in more detail.

**Authors Response:** We have added the details on the accuracy and land use classes in section 2.2 and added a reference paper by Msigwa et al. (2019) which provides more details about the land use classification. The explanation on the trajectories are found in section 2.3.

5) Innovative aspects of your research should be highlighted and presented against the state-of-the-art.

**Authors Response:** We agree with the reviewer that the state of art of the manuscript is not clear, we will include more explanation to our specific innovation. For example we have explained our innovation as the use of agro-hydrological model such as SWAT+ to representing blue and green ET for different cropping seasons (represented by trajectory with time and space) and the use of remote sensing ET to evaluate the simulated ET from SWAT+. Line 369-372

6) Proof-reading by a native speaker would be helpful. I suggested some changes, but there are certainly more sentences that need to be improved.

**Authors Response:** We have reviewed  and edited the  revised manuscript.

**Line specific comments:**

l.9-10: Please clarify and unify terms: cropping cycle, cropping seasons

**Authors Response:** Cropping cycle and cropping seasons were used interchangeably. We have rectified the issue by sticking with cropping seasons instead of cropping cycle.

l.11: 'In most agro-hydrological model applications such as SWAT+ in Africa, only one cropping season per year is represented.' This is indeed surprising. Please see also my comment on l. 56 and l. 72-73.

**Authors Response:** We have deleted the statement and added a new statement "most studies for mapping the blue and green water with agro-hydrological models such as SWAT do not capture these cropping seasons".

l.14: Better focus on the topic of this paper in the abstract: 'This study builds upon earlier research that proposed an approach on how to incorporate seasonal land use dynamics in the SWAT+ model but mainly focused on the temporal pattern of LAI and tested the approach in a small catchment (240 km2).'

**Authors Response**: The focus of the study has change, We have modified the introduction

l.20: suggest to change to: , remote sensing estimates, resulting in a higher performance' remove ,than default'

**Authors Response:** We have removed "than default" in the revised manuscript

l.22-23: Please improve the language and strengthen conclusion

**Authors Response:** We agree with the reviewer, the conclusion has been improved.

l.30 suggest ‚at the' instead of ‚per'

**Authors Response:** The introduction was changed hence the sentence is not there.

l.36: I believe these are studies that have implemented land-use dynamics. In this case, ‚few' is misleading, suggest to say ‚A few…'

**Authors Response:** Comment is well taken. We have change places we wrote "Few" to "A few"

l.44-45: Please clarify, what you mean with ‚implemented seasonal land-use dynamic in SWAT and SWAT+ through land-use trajectories, and not land-cover classes." As I understand it, a trajectory is also a change of land-use and land-cover classes. So that the meaning of the sentence is not clear to me.

**Authors Response:** The sentences were deleted and rephrased for clarity.

l.56: AfricaN basins

**Authors Response:** Africa has been changed to African

l.56 and 72-73:‚…typically not represent different cropping seasons' and  'Although the SWAT (+) model is capable of representing multiple cropping seasons, this is rarely implemented.'

I agree with you, that it is important to represent different cropping seasons. But please reflect that seasonal crop rotations can be depicted with SWAT and that has been done in the past in study areas with a strong seasonality, e.g. typically in India. Please find 3 example studies below. For these implementations the seasonal changes within one year is however always the same. Would it be possible to go beyond that with your methodology? Do you account for all possible combinations of seasonal crop rotations in space? Please highlight the innovation in your research.

Garg, K.K., Bharati, L., Gaur, A., George, B., Acharya, S., Jella, K. and Narasimhan, B. (2012), Spatial mapping of agricultural water productivity using the swat model in the Upper Bhima catchment, India. Irrig. and Drain., 61: 60-79. https://doi.org/10.1002/ird.618

Narsimlu, B., Gosain, A.K. & Chahar, B.R. Assessment of Future Climate Change Impacts on Water Resources of Upper Sind River Basin, India Using SWAT Model. Water Resour Manage 27, 3647–3662 (2013). https://doi.org/10.1007/s11269-013-0371-7

Wagner, P. D., Kumar, S., and Schneider, K.: An assessment of land use change impacts on the water resources of the Mula and Mutha Rivers catchment upstream of Pune, India, Hydrol. Earth Syst. Sci., 17, 2233–2246, https://doi.org/10.5194/hess-17-2233-2013, 2013.

**Authors Response:** Thank you for your comment. Yes we can go beyond these studies with our innovation since we are representing the trajectories within a year which include changes in time and space. We see an example of study in Pangani by Ndomba et al. 2008, the cropping seasons were not implemented because land use classification in the tropics is challenging as indicated by unclearly defined land use practices in this study area (Ndomba et al., 2008).

The paper by Merriman et al. (2019) is another example where the crop rotation has been implemented in details and where our innovation could be adopted. The study has shown crop rotation in detail e.g. "one crop cover, changing from corn silage to cereal rye". However, this method might not apply in most of the tropical African catchments like our catchment, because one crop cover may have different crop rotation practices. Hence, we need to know the location (having a specific rotation). For example, we have indicated that the same rainfed maize crop cover may change in one field and form a trajectory CORN→TOMA→TOMA (rainfed maize to tomato to tomato land use trajectory) or change from CORN →TOMA →BSVG (rainfed maize to tomato to sparse vegetation). Thus, it is very crucial to use the changes in seasonal land use to represent these changes in space and time.

l.74: 'By default, SWAT simulates a single growing cycle every year.' This is true, but it can be argued that the modeler should adjust the default, if the default is not applicable.

**Authors Response:** These sentences were omitted and rephrased since the introduction changed.

l.80-83: Please outline stronger what the new contribution of this paper is. If it building on earlier findings is fine, but this could also be outlined in the methods section.

**Authors Response:** Thank you for your comment. This is well noted. Our innovation is that we use an agro-hydrological model (SWAT+) to represent blue and green ET for different cropping seasons (represented by trajectory with time and space). Additionally, we used validated remote sensing ET to evaluate the simulated ET from SWAT+.

We have restructured the introduction and focused more on the blue and green ET. We have included the detailed explanation of our innovation in the revised manuscript.

l.80-92: Suggest to shorten the paragraph to the aims. Please move the methodological details to the methods section.

**Authors Response:** We have deleted the methodological details of the last paragraph in the introduction.

l.95 As there has been SWAT research on the Pangani basin, I would suggest to relate your research (literature review + findings) to it. See e.g.:

Notter, B., Hurni, H., Wiesmann, U., and Abbaspour, K. C.: Modelling water provision as an ecosystem service in a large East African river basin, Hydrol. Earth Syst. Sci., 16, 69–86, https://doi.org/10.5194/hess-16-69-2012, 2012.

**Authors Response:** The comment is well taken. We acknowledge related works using SWAT in the Pangani basin. However, we could not relate with the study by Notter, since the study didn't talk about the blue and green ET which is the focus of this study.

Fig. 1: Inset map is not readable. Please revise.

**Authors Response:** We have revised figure 1.

Fig. 2: It would be preferable to show a 30 year average of rainfall to depict the climate, if data is available. The authors state that there was at least data available for 2006-2013, l.122. Certainly, a longer period would be better. This figure is also depicted in Msigwa et al. 2019. Please, make sure that there are no copyright issues. You may include temperature to provide a bit more information here.

**Authors Response:** We revised the figure to include temperature, but we couldn't analyse a 30-year average due to missing data in most stations.

l.115: Please add which DEM was used, not only the source for download. SRTM?

**Authors Response:** The source of DEM has been added to the revised manuscript.

l.123-126: As the entire paper relies on the accuracy of these land use maps, you need to provide classification accuracies here. I would suggest to show at least overall accuracy and the range of user accuracies for the different land use classes. Please also state which and how many classes have been identified and which classification algorithm was applied.

**Authors Response:** The information on the classes accuracies have be included in the revised manuscript in section 2.2.

l.127: 'For instance,…' One example is not sufficient. Either provide the setup information for all land use classes or refer the reader to a publication where you have shown that.

**Authors Response:** We have referred the reader to the additional setup information in the appendix B, Table 3B within the revised manuscript.

l.136: Full stop missing

**Authors Response.** Thank you, we have include the full stop.

l.145-147: Sentence and reasoning not clear to me. Bananas and coffee should probably not change within a year. Did they in the trajectory analysis? If so, how would you explain that? Also, how would you parameterize a combined class of coffee and bananas? Please clarify.

**Authors Response:** We didn't implement the trajectory analysis for banana and coffee. However, in the catchment we have farms that change from only banana and coffee land use to banana, coffee and maize land use because they plant maize during the rainy season. This analysis was explained in the previous paper by Msigwa et al. (2019).

Figure 3: While this map provides a good first overview, regarding the topic of the paper, I think it is necessary to show the different land use trajectories in more detail.

**Authors Response:** The comment is well taken, we have revised the map and provide different land use trajectories in more detail.

l.164: Otherwise spelled as 'sub-basins', please unify.

**Authors Response:** Sub Basin have been changed to sub-basin in the whole revised manuscript.

l.170: Are you using the option to grow two or more crops at the same time? If yes, this should be highlighted, if not, why mention this?

**Authors Response:** We did not grow one crop at the same time. We have removed the highlight as suggested by the reviewer.

l.175-176: suggest to revise to 'limited amount of input data'

**Authors Response:** We removed dataset and replace with data.

l.177: 'rather than using remote sensing climate data' Sentence not clear, please clarify.

**Authors Response:** We have rephrased the statement in the revised manuscript.

l.181: Table 1B+2B do not show 40 trajectories, please clarify. Also, some of the trajectories seem to be no real rotations, e.g. "indn CORN-BSVG-BSVG", seems to be a single crop corn in one cropping season and no cropping in the other seasons. I think it should be highlighted which of these trajectories describe real crop rotations and which are only single crops, which could probably be well represented by a model without a seasonal representation of crops.

**Authors Response:** The comment is well taken. We have added table 2B in Appendix B to represent trajectories with real crop rotation.

Table 1: I would suggest to write 2-3 sentences to explain the shown management file highlighting the capabilities, e.g. tomato and soy bean are grown on the same field. Suggest to delete white

space. Moreover, if you have tomato and soy bean on one field, how was that derived in the land use classification? And if this was a class for itself, how good was the classification performance?

**Authors Response:** The reviewer comment is well taken. We have included a detailed explanation of the management file with the implementations. The tomato and soy bean were grown on one field at different times, the land use class was term as "irrigated mixed crop". Line 200-201

l.217: I cannot find the source 'IHE Delft, 2020' in the reference section.

**Authors Response:** The references have be reviewed and IHE Delft 2020 has been included in a revised manuscript.

l.239: 'statistical matrices'?

**Authors Response:** Sentence was deleted

l.213-239 The Model Evaluation section needs a thorough revision, please address the following points:

**Authors Response:** We have revised the section and add all the details suggested by the reviewer in the revised manuscript.

1) Setup of the two models: Which land use map was used for the static model?

**Authors Response:** The March land use was used for static model. We have explained this in the revised manuscript. Line 208.

2) Calibration approach? Did you calibrate your models? How did you do that and did you do this separately for the static and dynamic model?

**Authors Response:** We did not calibrate any model static or dynamic model. Our aim was to improve the spatial distribution of blue and green water consumption and not discharge simulation. We compared the static and dynamic model in default conditions. This approach allowed us to compare model results in default parameter conditions, considering parameter calibrations vary with different catchments. Nkwasa et al., (2020) also suggested that improved representation of

crop and agricultural land use processes should precede any model calibration efforts. Thus, we evaluated the simulated ET by comparing with the remote sensing ET.

3) It seems as if the model performance is solely evaluated with ET. This needs a better justification and explanation. What about the discharge data described in the methods section? Please provide more information on the ET data used for calibration (?) and validation. What exactly was compared? Basin values, sub-basin values, grid values? If that has been carried out in a previous study, you may also refer to that study for details, but you need to provide the reader with the main information that is necessary to evaluate the performance of your model.

**Authors Response:** The model performance was evaluated by the ET only. Hence, it was unnecessary to include the discharge data that was not used in the description. This has been omitted. Also, the description of how we compared and evaluated the remote sensing ET with the model simulated ET has been added in the revised manuscript and reference to a paper for further reading. Line 227-235.

See also the following HESS paper on SWAT modeling with ET data in Africa:

Odusanya, A. E., Mehdi, B., Schürz, C., Oke, A. O., Awokola, O. S., Awomeso, J. A., Adejuwon, J. O., and Schulz, K.: Multi-site calibration and validation of SWAT with satellite-based evapotranspiration in a data-sparse catchment in southwestern Nigeria, Hydrol. Earth Syst. Sci., 23, 1113–1144, https://doi.org/10.5194/hess-23-1113-2019, 2019.

4) Actually the indices that were applied are well known. I would suggest to rather focus on explaining the calibration and validation strategy and do not explain the indices in such detail.

**Authors Response:** The comment is well taken. An explanation on the indices has been omitted in the revised manuscript and more information on how we evaluated the ET has been added.

5) For which period was the model run?

**Authors Response:** The model run from a period of 2006 to 2013. The years 2006 and 2007 were used as a warming period. Details have been added in the revised manuscript (line 221).

l.253: Verb missing

**Authors Response:** The sentence has been revised.

l.260-262: Please explain and clarify, sentence not clear to me.

**Authors Response:** The sentence has been paraphrased and made clear in the revised manuscript.

l.266-268: Please revise sentence and check grammar.

**Authors Response:** The comment is well taken and we have revised the sentence.

Fig. 4: How come that the static ET peaks are some times higher than the dynamic ones? I would have assumed that dynamic ET =static ET for the period in which both have the same crop and that for all other seasons dynamic ET > static ET. As detailed and required information on how the static land use was implemented (and differs from the dynamic land use) is missing (see previous comment), it is hard to understand these differences.

**Authors Response:** We agree with the reviewer comment, we have revised the model setup in section 2.6. The model setup for static used a March land use map with only 14 land use classes, while the dynamic model used a land use map with 40 trajectories. Hence, the changes in the ET might be due to the different land use maps yielding different number of HRUs. A clear explanation will be added in the revised manuscript.

l.279: Suggest 'A notable difference…'

**Authors Response:** Comment is taken. We have change "The" to "A"

l.281: Please define what you refer to as 'mass balance in percentage'

**Authors Response:** Mass balance was meant to say change in soil water balance in the model. In the revised manuscript we have modified to include "change in soil water balance".

l.286-292 and Fig.5: How do you explain the strong differences for the areas that show a high satellite ET? Even the dynamic model underestimates these considerably.

**Authors Response:** The ET from dynamic could not reach maximum satellite ET because the satellite ET estimates also have uncertainties in the mountainous areas because of the presence of cloud cover. There are no observation data in these areas that we can validate with.

l.293-294: It is hard to follow the line of argumentation here. Looking at Figure 5 I see most pronounced changes between static and dynamic implementation at the Northern border of the catchment. But when I look at Fig. 2, these are not areas with trajectories. Please explain these differences. I would expect that all areas with no trajectories show the same ET value in both models.

**Authors Response:** Thank you for the comment. We have re-run the model and modify the parameters of the static land use to be the same for both models and hence ET is the same. We have revised the maps and they have change significantly.

l.295-297: Please clarify the following sentence: "Likewise, the changes seen in the high land areas of irrigated banana and coffee and the forested areas might be due to the increase in the number of HRUs in the dynamic SWAT+ model that contributed to the more accurate results." Why do HRU numbers change? Again the implementation differences between static and dynamic scenario are not clear. From a methodological point of view, I would not expect changes in the number of HRUs. For your study aims you need to make sure that you minimize any other impact (e.g. differences in model structures) to really deduce the impact of your seasonal land use change implementation.

**Authors Response:** The reviewers' comment is well taken. We have revised the explanation on the implementation of static and then dynamic models in section 2.7. The number of HRUs will be different because the input land use maps have different number of land uses classes. In the dynamic model the land use map (trajectory map) had 40 land use classes while the static land use map had 14 land use classes. However, we maintain the same number of sub-basins to try and minimize the differences in model structures.

l.308-309: Please improve language ‚for annual (Figure 6) and from 2008 to 2013.'

**Authors Response:** The caption language have been improved for figure 6.

l.320: As mentioned earlier: Please include a land use (trajectory) map, I cannot see where sugarcane is located. The reader must be able to follow and verify your conclusions.

**Authors Response:** The point is well taken. We have revised the land use map and include the trajectories.

l.324-335: These methods have not been explained. If you want to show these here, you need to include them in the methods section. It also looks as if some data from a forthcoming publication is shown. Please specify if you refer to the data or to the methods with the reference. See also my general comment on the two topics covered in this manuscript.

**Authors Response:** The comment is well taken. We have included an explanation of the methods for estimating blue and green ET that we have compared with the SWAT. We have also specify that we have used the data from a previous paper (Msigwa et al., 2021) on the revised manuscript.

Msigwa, A., Komakech, H. C., Salvadore, E., Seyoum, S., Mul, M. L., & van Griensven, A. (2021). Comparison of blue and green water fluxes for different land use classes in a semi-arid cultivated catchment using remote sensing. *Journal of Hydrology: Regional Studies*, *36*, 100860.

l.335: Forthcoming Msigwa et al. 2020 paper is not available in the reference section

**Authors Response:** Thank you for the comment we have included the reference in the reference section of the revised manuscript.

l.340: Please be more careful with this statement ‚none of these studies represented seasonal dynamics'. As outlined above, there are a number of studies that have incorporated seasonal crop rotations in India and possibly also elsewhere. They might not have compared the effect to a static model, but they still implemented them. Please highlight what the advantage of your approach is. One example might be the spatial representation of trajectories.

**Authors Response:** The comment is well taken we will revise the statement and add that our approach goes further to represent the spatial location of these seasonal changes.

l.342: Typo: You did show that, didn't you?

**Authors Response:** We meant the study by Nkwasa et al. 2020 didn't show how the seasonal land-use dynamic improved water balance component such as ET but not our current study. We will revise the sentence to make it clear in the revised manuscript.

l.349-352: Please also discuss and explain, why static and dynamic ET do not match for static land use areas and why your ET estimate never reaches the maximum satellite ET.

**Authors Response:** The comment is well taken. We will include the explanation in detail in the revised manuscript.

ET from static and dynamic model could not much in most areas due to the correction of the land use trajectories as explained earlier. However, this is a very important observation we will run the models again to have to confirm our argument.

The ET from dynamic could not reach maximum satellite ET because the satellite ET estimates also have uncertainties in the mountainous areas because of the presence of cloud cover. There are no observation data in these areas that we can validate with (Msigwa et al., 2021).

l.355: Forthcoming Msigwa et al. 2021 paper is not available in the reference section

**Authors Response:** Thank you for the comment. We will include the reference in the reference section of the revised manuscript.

l.365: What about the uncertainties of the land use maps and the associated trajectories as well as their impact on hydrology? Mostly it is hard to assess land use with multi-spectral satellite data in all seasons due to cloud cover (in the rainy season). How did you deal with this? And what does this mean for the transferability of your methodology?

**Authors Response:** We agree with the reviewer's comment. There were uncertainty associated with the trajectories for example unrealistic trajectories like change from crop to forest then crop again. These types of trajectories were corrected and reclassified. We will include these details in the discussion section.

We only used the images with less than 10% of the cloud cover. We also mask the clouds by replacing with the proper land use either using our field survey information or previous land use map by Kiptala et al. (2013). This approach had some effects since we were not able to capture the peak wet seasons where the crops have fully grown. We have include this limitation in the discussion section of the revised manuscript.

l.338: Please include a discussion of model performance in the discussion section.

**Authors Response**. We have revised the discussion part line 375-378 to include models performance.

l.385: ‚blue water amount is in line with previous studies' Not sure to which section the authors refer here and to which studies. Please clarify.

**Authors Response:** The reviewer's comment is well taken. We have revised the sentence and added the reference.

---

## Author Comment (AC7)

**Reply to Referee 3**

**Overview**: We would like to thank the reviewer for his/her dedication in reviewing the manuscript. We are also thankful for his/ her consideration and constructive suggestions and comments that have improve the revised manuscript.

Note: The lines are mentioned following simple makeup

**Reviewer summary**: The paper "Representation of seasonal land-use dynamics in SWAT+ for improved assessment of blue and green water consumption" reports an application of the SWAT+ model in Africa. The authors implemented seasonal dynamic land-use in SWAT+ in order to improve vegetation growth simulation and to obtain more realistic temporal patterns of the blue and green water consumption from simulated evapotranspiration. Results of the simulations (static and dynamic seasonal land use) in terms of ET were compared to the ET values estimated by using remote sensing. The authors concluded that the seasonal land-use dynamic approach produces better ET results, which provides better estimations of blue and green water.

General comments:

**Reviewer comment**: The paper is very similar to a previous work that has been published in 2020 by the same research group (Nkwasa et al., 2020). The latter paper showed a better performance of the SWAT+ model by using the seasonal land use dynamic (ET at HRU level after the implementation of trajectories in SWAT+ model was compared to the default SWAT+ model). In addition, in 2019 the authors published a study carried out in the same basin applying SWAT+ with dynamic land use and the authors concluded that detailed seasonal land use maps are essential for quantifying annual irrigation water use of catchment areas. For these reasons, it seems difficult to find the novelty of the present paper. Hence, I invite the author to revise the introduction in order to better focus on the advancement of knowledge proposed in this study. Taking into account that the methodological approach (seasonal land-use dynamics in SWAT+) has already been published, the authors should better focus on the green and blue waters.

I suggest major revisions, the current version cannot be published in HESS.

**Authors Response**: The introduction and subsequent sections of the manuscript have been revised to focus more on the blue and green water.

To make a clear distinction with earlier studies, we have first introduced blue and green water with definitions, and their relevance. However, mapping the blue and green water is possible with agro-hydrological models such as SWAT+ but then they need a better representation of the seasonality/cropping seasons. Hence, this can be done by the trajectory approach in SWAT+ as suggested by Nkwasa et al. (2020). The innovation is that we use an agro-hydrological model (SWAT+) to evaluate blue and green water by implementing the seasonal land-use dynamic. Additionally, we use remote sensing ET for evaluation of the simulated model ET.

Methodology:

**Reviewer comment**: a better description of the remote sensing ET evaluation is needed, the reference IHE Delft, 2020 is not listed.

**Authors Response:** The reviewer comment is well taken. We added details on the remote sensing ET (line 228-235) and also included the 'IHE Delft; 2020' in the reference list.

**Reviewer comment**: More details on irrigation are needed, for instance, the source of water for irrigation (i.e. from the river, shallow aquifer, etc). Analyzing table 1, it seems that the option auto-irrigation was used. Please explain it. Did the author compare the amount of auto-irrigation to the actual irrigation (data provided by farmers)?

**Authors Response:** In the revised manuscript, we included the details of irrigation such as the source of water (river) (line 200-202). We failed to evaluate the auto- irrigation in comparison with the actual irrigation through the information gathered in the field, we have limited data of measured water applied in the field.

**Reviewer comment**: In my opinion, the equations and description of the RMSE, PBIAS, and NSE are not necessary.

**Authors Response:** We agree with the reviewer comment. We have deleted the description of the RMSE, PBIAS and NSE in the revised manuscript.

**Reviewer comment**: For which period was the model run?

**Authors Response:** The model run from a period of 2006 to 2013. The years 2006 and 2007 were used as a warming period.

**Reviewer comment**: Figure 2 has already been reported in Msigwa et al., 2019 and for this reason, I suggest do not report it here.

**Authors Response:** We agree with the comment. However, we have revised the figure and added new information of temperature. According to reviewer 2 comments.

**Reviewer comment**: I suggest adding a new map in figure 3 with the land use (static land use).

**Authors Response:** The comment is well taken. We have added information on both the trajectory and static land use maps.

**Reviewer comment**: Calibration needs a better presentation. It seems that the calibration was performed for the static and dynamic approach, please show the calibrated parameters in both simulations. A table with calibrated parameters for both simulations is expected. What about validation?

**Authors Response:** We did not calibrate the SWAT+ model, our aim was mainly to improve the spatial distribution of blue and green water consumption and not discharge simulation. We evaluated the simulated ET by comparing with the remote sensing ET. A detailed explanation has been added in section 2.7.

Result section:

**Reviewer comment**: Methods reported in Lines from 324 to 335 are not reported in the "Material and methods" section. What is the aim to show them in Figure 8? In my opinion, this section should be eliminated.

**Authors Response:** The comment is well taken. We have added a brief explanation of the methods for estimating blue and green ET that we have compared with the SWAT+ model results. The section 2.9 is added.

The aim of comparing blue and green ET estimates was to provide evidence that the blue ET estimates from dynamic SWAT+ model show no statistical difference with the blue ET estimated using remote sensing with Van Eekelen method.

**Reviewer comment**: Line 335. Caption Figure 8. Msigwa et al. 2020 is not reported in the references. Is this reference the same as that reported in Line 355 Msigwa et al. 2021 (missed in the reference)?

**Authors Response:** Thank you for the comment we have included the references in the reference section of the revised manuscript.

Discussions:

**Reviewer comment**: Innovative aspects of your research should be highlighted and presented against the state-of-the-art. The authors reported (LINE 355) that blue and green ET estimates from SWAT+ for the mixed crop land-use show no significant difference in the values from the two methods (EK and SWB) assessed in the upcoming paper by Msigwa et al., (2021). This is not the aim of the present paper.

**Authors Response:** The paper now focuses on the blue and green ET, hence it is relevant for comparison of blue ET from other method.

**Reviewer comment**: Please discuss the difference between Figure 5b and fig. 5c and their comparison with figure 7. I did not understand why static and dynamic ET do not match for static land use areas. In the upper right corner, figure5b shows the green areas in correspondence with the static land use (see fig 3). The large difference is difficult to explain with a different number of HRUs. In addition, a large difference remains between dynamic and satellite ET (Fig 5a and 5b) that needs to be explained.

**Authors Response:** We agree with the reviewer comment that the ET for static land uses needs to be same. We have reviewed the model setups and rectified our simulations.

The number of HRUs will be different because the input land use maps have different number of land uses classes. Dynamic model the land use map (trajectory map) had 40 land use classes while the static land use map had 14 land use classes. However, the same number of subbasins is maintained to try and preserve a similar model structure.

The ET from the dynamic model setup could not reach maximum satellite ET because the satellite ET estimates also have uncertainties in the mountainous areas due to the presence of cloud cover. There are no observation data in these areas that we can validate with. We also used the rainfall ground stations for ET simulation while the remote sensing use the energy balance models. This explanation is added in the discussion section (Line 375-377)

**Reviewer comment**: Please discuss differences in water balance components between static and dynamic scenarios.

**Authors Response:** The reviewer's comment is well taken. Explanation of differences in water balance components between static and dynamic has been included in the revised manuscript line 177-178.

**Reviewer comment**: Please discuss the limit of the present study.

**Authors Response** We have discussed the limitation of our study. (Line 385-394)

**Reviewer comment**: Conclusions need to be improved.

**Authors Response:** We agree with the reviewer's comment. We have revised and improved the conclusion section.

**Reviewer comment**: The authors reported, "The maps with calculated blue water use from the dynamic SWAT+ model correspond to the known irrigated area and the calculated blue water amount is in line with previous studies". The first assertion is obvious since the authors set the irrigation in that areas. The second was expected since the authors refer to their previous papers.

**Authors Response:** We have rephrased the sentence.

**Reviewer comment**: The paper needs to be carefully checked for typing errors (see some of them have been highlighted in the file enclosed)

**Authors Response:** We have implemented all the corrections and checked for other typing errors in the revised manuscript.

---

## Author Response (AR2)

**Reply to referee 2**

**Overview**: We would like to thank the reviewer for his/her effort and time in reviewing the manuscript. We are also thankful for his/her constructive suggestions and comments in the third review.

Note: The lines are according to simple markup.

Dear authors,

Thank you very much for the revision of your manuscript. The paper has gained a lot through the revision. As the land use setup is now clearer, a major point that should be addressed is the comparison to the baseline (static) model. Even though I understand that you want to show the improvement of model simulation due to the implemented changes, the baseline model should also have a valid model representation (see comment on section 2.6). In the following I list a few points that should be dealt with or clarified before publication.

Response to earlier comments:

Earlier comment: l.95 As there has been SWAT research on the Pangani basin, I would suggest to relate your research (literature review + findings) to it. See e.g.:

Notter, B., Hurni, H., Wiesmann, U., and Abbaspour, K. C.: Modelling water provision as an ecosystem service in a large East African river basin, Hydrol. Earth Syst. Sci., 16, 69–86, https://doi.org/10.5194/hess-16-69-2012, 2012.

"Authors Response: The comment is well taken. We acknowledge related works using SWAT in the Pangani basin. However, we could not relate with the study by Notter, since the study didn't talk about the blue and green ET which is the focus of this study."

New comment: It is good scientific practice to review the work that has been done in your catchment – particularly if the same model has been used. Even if they do not deal with your particular focus area. I would therefore strongly recommend to look at the paper by Notter et al. 2012. How did they deal with the land use configuration in the model? This will strengthen your paper and as a reader I would like to know how your approach differs from what has been done in the area.

**Response:** Thank you for your comment, we have referenced the study by Notter et al. 2012 under the section 2.2 where we explained the LULC maps used in the study (line 128-130). Also, where we describe the SWAT model line 194-195.

Earlier comment: l.115: Please add which DEM was used, not only the source for download. SRTM?

Authors Response: The source of DEM has been added to the revised manuscript

New comment: Could you please add which data was used? USGS provides six different products.

**Response:** Thank you for your comment the product name SRTM was added in the line 119 to 120.

Earlier comment: Table 1: I would suggest to write 2-3 sentences to explain the shown management file highlighting the capabilities, e.g. tomato and soy bean are grown on the same field. Suggest to delete white space. Moreover, if you have tomato and soy bean on one field, how was that derived in the land use classification? And if this was a class for itself, how good was the classification performance?

Authors Response: The reviewer comment is well taken. We have included a detailed explanation of the management file with the implementations. The tomato and soy bean were grown on one field at different times, the land use class was term as "irrigated mixed crop". Line 200-201

New comment: As far as I understand Table 1, soybean is planted on 1 July and harvested on 1 October; tomato is planted on 20 August and harvested on 20 October. So these plants are grown at the same time on the same field. Please clarify.

**Response:** The reviewer comment is well taken. It is true that the plants (soya beans and tomato) are grown at the same period. The land use class for the beans and tomato was irrigated mixed crops because mostly the beans and tomato are planted on the same field. Line 276-278 explanation is added.

Earlier comment: 5) For which period was the model run?

Authors Response: The model run from a period of 2006 to 2013. The years 2006 and 2007 were used as a warming period. Details have been added in the revised manuscript (line 221).

New comment: The satellite derived ET is from the same time span (2006-2013), right? Please clarify.

**Response:** The reviewer comment is noted. The satellite ET was from year 2008 to 2013 because 2006 and 2007 was a warming period. We added this statement to the model evaluation section 2.7 line 227 "*for the same simulation period from 2008 to 2013*"

Earlier comment: Fig. 4: How come that the static ET peaks are some times higher than the dynamic ones? I would have assumed that dynamic ET =static ET for the period in which both have the same crop and that for all other seasons dynamic ET > static ET. As detailed and required information on how the static land use was implemented (and differs from the dynamic land use) is missing (see previous comment), it is hard to understand these differences.

Authors Response: We agree with the reviewer comment, we have revised the model setup in section 2.6. The model setup for static used a March land use map with only 14 land use classes, while the

dynamic model used a land use map with 40 trajectories. Hence, the changes in the ET might be due to the different land use maps yielding different number of HRUs. A clear explanation will be added in the revised manuscript.

New comment: Actually, I was not able to find this explanation in the changed manuscript.

**Response:** Thank you for your observation, new paragraph has been added to the discussion section from line 402 to 408.

Line specific comments (line numbers with regard to track-change document):

Please check language, e.g. in:

    a) l.28-30. "… the SWAT+ blue and green ET are similar to the van Eekelen method…" Better split up this long sentence and formulate more clearly.

**Response:** Thank you for the observation, the sentence was spilt into two sentences

    b) l.479: "are significant difference from"

**Response:** The sentence was revised "significant difference" was changed into "significantly different"

    c) l.486: "ET from dynamic could not reach…"

**Response:** The sentence was revised '' The ET from dynamic land-use setup could not reach maximum satellite ET because the satellite ET estimates also have uncertainties in the mountainous areas because of the presence of cloud cover.''

Abstract: a) Term blue and green ET is introduced in l.340 – if you use it early (abstract) it should be introduced there as well, b) Use of citations is usually discouraged in the abstract

**Response:** The reviewer comment is taken; the term blue and green ET was introduced in the abstract section.

l.42: Sentence not precise. I think only the abstracted water is used for irrigation, the rainwater is used by the plant, but not for irrigation. Please modify the sentence

**Response:** The reviewer comment is taken the line 27 the word "irrigation" was deleted.

l.105: Please check out the work of Jain et al. 2013, they consider seasons for deriving cropping intensity. So, you may include this to show that there has been related work.

Jain, M., Mondal, P., DeFries, R.S., Small, C. and Galford, G.L., 2013. Mapping cropping intensity of smallholder farms: A comparison of methods using multiple sensors. Remote Sensing of Environment, 134, pp.210-223.

**Response:** Thank for the reference paper. We have added the paper to our reference and cited as one of the works that at least assess the cropping intensity, in the introduction section line 75-77

Fig. 1+5+8: Legend should show that colors refer to ranges and not to one specific number

**Response:** The figures' color range were specified.

Fig. 3: The 10 classes should be explained in the legend. Could you relate this to the table of rotations given in the appendix?

**Response:** The classes meaning were explained.

Fig.6: Are the colors actually referring to one number or to a range?

**Response:** The figure was changed, and colors refers now to ranges.

2.5. a) Please highlight that the trajectories differ in space and that you thereby go beyond traditional approaches that use fixed crop rotations that are based on e.g. agricultural statistics or expert knowledge.

**Response:** The explanation was added under line 198 to 199. "These trajectories differ from the traditional approach as they not only use the agricultural statics but use land use maps to define the space".

2.5 b) I would suggest to reconsider the term 'trajectories'. In line 199 you provide the following definition 'In this study, we extended the meaning of land-use trajectories from 'land-use change' to 'seasonal succession of land-use types for a given sample unit (pixel) with more than two observations at different times''. I would argue that most readers would associate a land use trajectory with changes beyond one year. I actually thought that you were also considering changes between years, but in fact, you are only considering seasonal changes. So, your trajectory is limited to one year. I think that the term 'seasonal land-use dynamics' (as used in the title) is probably more suitable. I would leave that to the authors, but in any case, clearly state that your dynamics or trajectories are limited to one year.

**Response:** In this study we adopted trajectory analysis method to assess the seasonal land use changes as explained in the paragraph 80 to 86. The term trajectory was to show the path taken by the cropping system in the catchment for example the rainfed maize to irrigated tomato to irrigated maize. We have used the same method in our previous study by Nkwasa et al. 2020. We also explain the terms in the methodology section line 146-149.

2.6 So for the static model, you only use the first crop mentioned in your dynamic schedule, right? Please add, how you exactly deal with this. Is the crop harvested at the same time as in the dynamic

schedule and the field is left bare after harvest? Or do you plant the static crop three times? My point is, that the static model should also have a valid representation of agriculture in the region. If you leave the fields bare during 2 of 3 seasons, it is not surprising that the dynamic model generated more ET. So, you cannot say, if your improvement is based on better information on seasonal crops or if it is based on a very basic model setup of the static model. The conclusions from your study would be much stronger if you compare your model to a reasonable static model representation.

**Response:** For the static model the March land use map had crops grown only in the rainy season from March to July and later the land is left bare. This is the normal practice of model implementation in the African catchment region

l.465-469: "However, none of these studies has represented the seasonal dynamics of land use within a single year." As some studies consider seasonal crop rotations, I would suggest to change to "However, none of these studies has represented the seasonal dynamics of land use within a single year in a spatially distributed manner."

**Response:** Thank you for your suggestions the line 379 to 380 was changed according to the reviewer's suggestion

**Reply to referee 3**

**Overview**:  We would like to thank the reviewer for his/her effort and valuable time in reviewing the manuscript. We are also thankful for the positive response after the second review.

Note: The lines are according to simple markup

Dear Editor,

The paper has been improved, the main critical issues has been addressed.

**Response:** Thank you for the positive comment

The current version of the paper requires the following minor revisions:

• 	Line 119. "Daily discharge records were obtained from PBWO"

I suggest to delete this sentence since the author did not calibrate the model for streamflow and they did not use these data.

**Response:** The comment is well noted, the line 119 was deleted

• 	Line 385. "Moreover, different data for estimating ET could lead to these differences"

Please explain "different data"

**Response:** The line 385 was modified from different data to different method and following the line 386 example of data used were added "*Climate ground stations (temperature, wind speed, relative humidity and solar radiation)*"

• Line 390. "These findings demonstrate the importance of the representation of seasonal land-use dynamic in modelling hydrological models when quantifying blue and green water consumption"

Please check the sentence. It is better "These findings demonstrate the importance of the representation of seasonal land-use dynamic in modelling blue and green water consumption"

**Response:** The comment is well noted, the sentence was modified as suggested by the reviewer.

---

## Author Response (AR3)

**Reply to Editor**

**Overview**:  We would like to thank the editor and reviewer for their comment to refine this manuscript.

Note: The lines are according to all markup.

Dear Authors,

I thank you for having added in your manuscript the clarifications based the further comments by the two referees, but I confess that I am not sure you have fully addressed the issue that Ref#1 (and I) underlined on the adequacy of the benchmarking static model.

"2.6 So for the static model, you only use the first crop mentioned in your dynamic schedule, right? Please add, how you exactly deal with this. Is the crop harvested at the same time as in the dynamic schedule and the field is left bare after harvest? Or do you plant the static crop three times? My point is, that the static model should also have a valid representation of agriculture in the region. If you leave the fields bare during 2 of 3 seasons, it is not surprising that the dynamic model generated more ET. So, you cannot say, if your improvement is based on better information on seasonal crops or if it is based on a very basic model setup of the static model. The conclusions from your study would be much stronger if you compare your model to a reasonable static model representation.
Response: For the static model the March land use map had crops grown only in the rainy season from March to July and later the land is left bare. This is the normal practice of model implementation in the African catchment region"

As Ref#1 highlighted, for including the impact of having more than one crop in the same year in a benchmark, simplified modelling scheme, maybe it would not be necessary the use of seasonal land-use dynamics, but (at least if the crop is the same), it may be possible simulating a static model with the same LCLU, where three consecutive growing seasons are foreseen? I am not a SWAT user, so I don't know if this would be feasible, but it would be a more challenging comparison.
If this kind of procedure is not possible and it is not have been done, so far, I would ask you to add in the revised paper a specific comment on that issue, explicitating such limitation of the comparison with the static approach. And you should also support the fact that the implementation you propose for the "default" static benchmark is indeed the "standard one" in the case study region, adding references to previous works.
I believe that the value of you work does not depend only on the comparison with the "static"

modelling, since you also include the validation with the remote sensing estimates, but more clarity on this issue is definitely needed.

Response: Thank you for your comment, yes, the static model used the one crop that is represented in the dynamic then the farm is left bare, we have added the explanation in the section 2.6 line 113-115. It is true that this is not a real representation of agricultural management in the catchment, but it is how most modelers in African catchment simulate SWAT model, we have added this explanation in the introduction from line 103 to 114. This was the one of the objectives of this paper to show that better representation of cropping seasons is needed, and we showed the effect of implementing seasonal dynamic.

The referee 1 comment, it is true that we can simulate in SWAT with one LULC map then do crop rotation later. However, the nature of the small-scale agricultural practices in tropical African catchment with an example of Kikuletwa catchment is that one crop is planted in one season and harvested, and then different crop is planted (different crop). Hence the seasonal land use is important so as to identify the location and type of crop. The practice in African catchment is not same as outside African catchments like Europe, because first of small-scale agriculture practice with different kind of crops from each farmer. Also, the fact that all farmers don't have a systematic schedule to follow. Hence the seasonal land use maps help to obtain the information of crop specific, agriculture management per location.

**Notification to the authors**:

I've just noticed that your figure 3a contains an aerial. Please check whether an appropriate copyright/image credit is required and add it either in the figure itself or in the figure caption. If you are the originator, you can just inform us.

Response: The image was obtained from the google earth, with permission to use, distribute but not for commercial purposes. However, the whole figure creation, the authors are the originators.

---

## Author Response (AR4)

**Editor comments**: dear authors,

if you cannot add the application with more growing seasons with the same crop, a clarification of the limitations of your approach fort he static benchmark in the text is sufficent, but such clarification is not clear from your reply.

The lines you cite:

"we have added the explanation in the section 2.6 line 113-115."

"we have added this explanation in the introduction from line 103 to 114."

do not match with the final version with no track-changes and the track-change version is not readable due to the many colors...

Please report the full text of the new phrases you have added in the reply letter and send a track-change version including only this last round of modification in respect to the version we had already revised.

Note the line are according to all mark up track

**Response:** We are sorry that the lines to our response could not match, we forgot to change after the paper came back that we needed to redo the figures in the manuscript.

From our previous response "the static model used the one crop that is represented in the dynamic then the farm is left bare, we have added the explanation in the section 2.6 line 236-237. It is true that this is not a real representation of agricultural management in the catchment, but it is how most modelers in African catchment simulate SWAT model, we have added this explanation in the introduction from line 84 to 87. This was the one of the objectives of this paper to show that better representation of cropping seasons is needed, and we showed the effect of implementing seasonal dynamic".